# Don't blame Dataset Shift! Shortcut Learning due to Gradients and Cross Entropy

**Aahlad Puli**[1]* **Lily Zhang** [2] **Yoav Wald** [2] **Rajesh Ranganath**[1,2,3]

[1]Department of Computer Science, New York University
[2]Center for Data Science, New York University
[3]Department of Population Health, Langone Health, New York University

## Abstract

Common explanations for shortcut learning assume that the shortcut improves prediction under the training distribution but not in the test distribution. Thus, models trained via the typical gradient-based optimization of cross-entropy, which we call default-ERM, utilize the shortcut. However, even when the stable feature determines the label in the training distribution and the shortcut does not provide any additional information, like in perception tasks, default-ERM still exhibits shortcut learning. Why are such solutions preferred when the loss for default-ERM can be driven to zero using the stable feature alone? By studying a linear perception task, we show that default-ERM's preference for maximizing the margin leads to models that depend more on the shortcut than the stable feature, even without overparameterization. This insight suggests that default-ERM's implicit inductive bias towards max-margin is unsuitable for perception tasks. Instead, we develop an inductive bias toward uniform margins and show that this bias guarantees dependence only on the perfect stable feature in the linear perception task. We develop loss functions that encourage uniform-margin solutions, called margin control (MARG-CTRL). MARG-CTRL mitigates shortcut learning on a variety of vision and language tasks, showing that better inductive biases can remove the need for expensive two-stage shortcut-mitigating methods in perception tasks.

## 1 Introduction

Shortcut learning is a phenomenon where a model learns to base its predictions on an unstable correlation, or *shortcut*, that does not hold across data distributions collected at different times and/or places [Geirhos et al., 2020]. A model that learns shortcuts can perform worse than random guessing in settings where the label's relationship with the shortcut feature changes [Koh et al., 2021, Puli et al., 2022]. Such drops in performance do not occur if the model depends on features whose relationship with the label does not change across settings; these are *stable* features.

Shortcut learning is well studied in cases where models that use both shortcut and stable features achieve lower loss than models that only use the stable feature [Arjovsky et al., 2019, Puli et al., 2022, Geirhos et al., 2020]. These works consider cases where the Bayes-optimal classifier — the training conditional distribution of the label given the covariates — depends on both stable and shortcut features. In such cases, shortcut learning occurs as the Bayes-optimal predictor is the target of standard supervised learning algorithms such as the one that minimizes the log-loss via gradient descent (GD), which we call default-ERM.

However, in many machine learning tasks, the stable feature perfectly predicts the label, i.e. a *perfect stable feature*. For example, in task of predicting hair color from images of celebrity faces in the

---

*[1]Corresponding email: `aahlad@nyu.edu`

37th Conference on Neural Information Processing Systems (NeurIPS 2023).

CelebA dataset [Sagawa et al., 2020a], the color of the hair in the image determines the label. This task is a perception task. In such classification tasks, the label is independent of the shortcut feature given the stable feature, and the Bayes-optimal predictor under the training distribution only depends on the stable feature. Default-ERM can learn this Bayes-optimal classifier which, by depending solely on the stable feature, also generalizes outside the training distribution. But in practice, default-ERM run on finite data yields models that depend on the shortcut and thus perform worse than chance outside the training distribution [Sagawa et al., 2020a, Liu et al., 2021, Zhang et al., 2022]. The question is, why does default-ERM prefer models that exploit the shortcut even when a model can achieve zero loss using the stable feature alone?

To understand preferences toward shortcuts, we study default-ERM on a linear perception task with a stable feature that determines the label and a shortcut feature that does not. The perfect linear stable feature means that data is linearly separable. This separability means that default-ERM-trained linear models classify in the same way as the minimum $\ell_2$-norm solution that has all margins greater than 1; the latter is commonly called max-margin classification [Soudry et al., 2018]. We prove that default-ERM's implicit inductive bias toward the max-margin solution is harmful in that default-ERM-trained linear models depend more on the shortcut than the stable feature. In fact, such dependence on the shortcut occurs even in the setting with fewer parameters in the linear model than data points, i.e. without overparameterization. These observations suggest that a max-margin inductive bias is unsuitable for perception tasks.

Next, we study inductive biases more suitable for perception tasks with perfect stable features. We first observe that predicting with the perfect stable feature alone achieves uniform margins on all samples. Formally, if the stable feature $s(\mathbf{x})$ determines the label $\mathbf{y}$ via a function $d$, $\mathbf{y} = d \circ s(\mathbf{x})$, one can achieve any positive $b$ as the margin on all samples simultaneously by predicting with $b \cdot d \circ s(\mathbf{x})$. We show that in the same setting without overparameterization where max-margin classification leads to shortcut learning, models that classify with uniform margins depend only on the stable feature.

Building on these observations, we identify alternative loss functions that are inductively biased toward uniform margins, which we call margin control (MARG-CTRL). We empirically demonstrate that MARG-CTRL mitigates shortcut learning on multiple vision and language tasks without the use of annotations of the shortcut feature in training. Further, MARG-CTRL performs on par or better than the more expensive two-stage shortcut-mitigating methods [Liu et al., 2021, Zhang et al., 2022]. We then introduce a more challenging setting where both training and validation shortcut annotations are unavailable, called the nuisance-free setting. In the nuisance-free setting, MARG-CTRL *always outperforms* default-ERM and the two-stage shortcut-mitigating methods. These empirical results suggest that simply incorporating inductive biases more suitable for perception tasks is sufficient to mitigate shortcuts.

## 2 Shortcut learning in perception tasks due to maximizing margins

**Setup.** We use $\mathbf{y}, \mathbf{z}, \mathbf{x}$ to denote the label, the shortcut feature, and the covariates respectively. We let the training and test distributions $(p_{tr}, p_{te})$ be members of a family of distributions indexed by $\rho$, $\mathcal{F} = \{p_\rho(\mathbf{y}, \mathbf{z}, \mathbf{x})\}_\rho$, such that the shortcut-label relationship $p_\rho(\mathbf{z}, \mathbf{y})$ changes over the family. Many common tasks in the spurious correlations literature have stable features $s(\mathbf{x})$ that are perfect, meaning that the label is a deterministic function $d$ of the stable feature: $\mathbf{y} = d \circ s(\mathbf{x})$. For example, in the Waterbirds task the bird's body determines the label and in the CelebA task, hair color determines the label [Sagawa et al., 2020a]. As $s(\mathbf{x})$ determines the label, it holds that $\mathbf{y} \perp\!\!\!\perp_{p_\rho} (\mathbf{x}, \mathbf{z}) \mid s(\mathbf{x})$. Then, the optimal predictor on the training distribution is optimal on all distributions in the family $\mathcal{F}$, regardless of the shortcut because

$$p_{tr}(\mathbf{y} \mid \mathbf{x}) = p_{tr}(\mathbf{y} \mid s(\mathbf{x})) = p_{te}(\mathbf{y} \mid s(\mathbf{x})) = p_{te}(\mathbf{y} \mid \mathbf{x}).$$

The most common procedure to train predictive models to approximate $p_{tr}(\mathbf{y} \mid \mathbf{x})$ is gradient-based optimization of cross-entropy (also called log-loss); we call this default-ERM. Default-ERM targets the Bayes-optimal predictor of the training distribution which, in tasks with perfect stable features, also performs optimally under the test distribution. However, despite targeting the predictor that does not depend on the shortcut, models built with default-ERM still rely on shortcut features that are often less predictive of the label and are unstable, i.e. vary across distributions [Geirhos et al., 2020, Puli et al., 2022]. We study default-ERM's preference for shortcuts in a data generating process (DGP) where both the shortcut and the perfect stable feature are linear functions of the covariates.

## 2.1 Shortcut learning in linear perception tasks

Let Rad be the uniform distribution over $\{1, -1\}$, $\mathcal{N}$ be the normal distribution, $d$ be the dimension of $\mathbf{x}$, and $\rho \in (0, 1), B > 1$ be scalar constants. The DGP for $p_\rho(\mathbf{y}, \mathbf{z}, \mathbf{x})$ is:

$$\mathbf{y} \sim \text{Rad}, \quad \mathbf{z} \sim \begin{cases} p_\rho(\mathbf{z} = y \mid \mathbf{y} = y) = \rho \\ p_\rho(\mathbf{z} = -y \mid \mathbf{y} = y) = (1 - \rho) \end{cases}, \quad \boldsymbol{\delta} \sim \mathcal{N}(0, \mathbf{I}^{d-2}), \quad \mathbf{x} = [B * \mathbf{z}, \mathbf{y}, \boldsymbol{\delta}]. \quad (1)$$

This DGP is set up to mirror the empirical evidence in the literature showing that shortcut features are typically learned first [Sagawa et al., 2020a]. The first dimension of $\mathbf{x}$, i.e. $\mathbf{x}_1$, is a shortcut that is correlated with $\mathbf{y}$ according to $\rho$. The factor $B$ in $\mathbf{x}_1$ scales up the gradients for parameters that interact with $\mathbf{x}_1$ in predictions. For large enough $B$, model dependence on the shortcut feature during default-ERM goes up faster than the stable feature [Idrissi et al., 2022].

The training distribution is $p_{tr} = p_{0.9}$ and the test distribution is one where the short-cut's relationship with the label is flipped $p_{te} = p_{0.1}$. Models achieve worse than random test accuracy ($50\%$) if they exploit the training shortcut relationship and the predicted class flips when the shortcut feature flips. We train with default-ERM which uses log-loss: on a data point $(\mathbf{x}, \mathbf{y})$ the log-loss is

$$\ell_{log}(\mathbf{y} f_\theta(\mathbf{x})) = \log\left[1 + \exp(-\mathbf{y} f_\theta(\mathbf{x}))\right].$$

With $d = 300$ and $B = 10$, we train a linear model on 1000 samples from the training distribution $p_{\rho=0.9}$, and evaluate on 1000 samples from $p_{\rho=0.1}$.

**Observations.** Figure 1a shows that when trained with default-ERM, the linear model does not do better than chance ($< 50\%$) on the test data even after $50,000$ epochs. So, even in the presence of the perfect feature $\mathbf{x}_2$, the model relies on other features like the shortcut $\mathbf{x}_1$. Since the final training loss is very small, on the order of $10^{-9}$, this result is not due to optimization being stuck in a local minima with high loss. **These observations indicate that, in the linear setting, gradient-based optimization with log-loss prefers models that depend more on the shortcut than the perfect stable feature.**

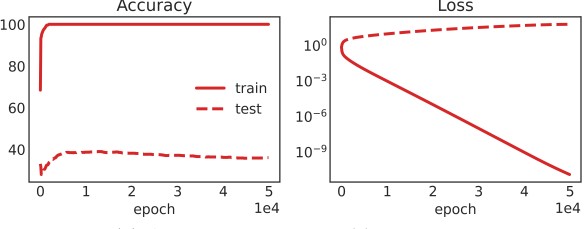

**(a)** Average accuracy and loss curves.

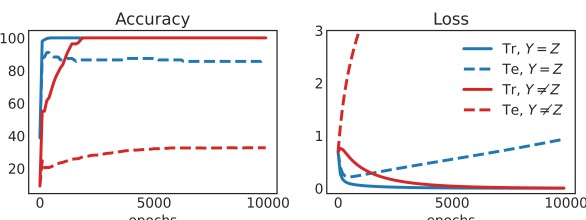

**(b)** Accuracy and loss on shortcut and leftover groups.

**Figure 1:** Accuracy and loss curves for training a linear model with default-ERM on 1000 training samples from $p_{0.9}$, with $B = 10, d = 300$ (see eq. (1)), and testing on $p_{0.1}$. **(a)** The model achieves $100\%$ train accuracy but $< 40\%$ test accuracy. **(b)** The learned model achieves high test accuracy ($\approx 90\%$) on the shortcut group and low test accuracy on the leftover group ($\approx 30\%$). Models that depend more on the stable feature than on the shortcut, achieve at least $50\%$ accuracy on both the shortcut and leftover groups. Hence the learned model exploits the shortcut to classify the shortcut group and overfits to the leftover group.

To better understand this preference we focus on the errors in specific groups in the data. Consider the classifier that only uses the shortcut $\mathbf{z}$ and makes the Bayes-optimal prediction w.r.t $p_{tr}$: $\arg\max_y p_{tr}(\mathbf{y} = y \mid \mathbf{z})$. We call instances that are classified correctly by this model the *shortcut* group, and the rest the *leftover* group. We use these terms for instances in the training set as well as the test set. In this experiment $\mathbf{y}$ is positively correlated with $\mathbf{z}$, hence the shortcut group consists of all instances with $\mathbf{y}^i = \mathbf{z}^i$ and the leftover group of those with $\mathbf{y}^i \neq \mathbf{z}^i$.

Figure 1b gives accuracy and loss curves on the shortcut and leftover groups for the first 10000 epochs. The test accuracy for the shortcut group hits $90\%$ while the leftover group test accuracy is $< 40\%$, meaning that the model exploits the shortcuts. Even though a model that relies solely on the shortcut misclassifies the leftover group, we see that the training loss of the learned model on this group approaches 0. The model drives down training loss in the leftover group by depending on noise, which results in larger test loss in the leftover group than the shortcut group. **Thus, fig. 1b demonstrates that the default-ERM-trained model classifies the training shortcut group by using the shortcut feature while overfitting to the training leftover group.**

Shortcut dependence like in fig. 1 occurs even with $\ell_2$-regularization and when training neural networks; see appendix B.1 and appendix B.4 respectively. Next, we analyze the failure mode in fig. 1, showing that the shortcut dependence is due to default-ERM's implicit bias to learn the max-margin classifier. Next, we study the failure mode in fig. 1 theoretically, showing that the shortcut dependence is due to default-ERM's inductive bias toward learning the max-margin classifier.

**Max-margin classifiers depend more on the the shortcut than the stable feature.** We consider training a linear model $f_\theta(\mathbf{x}) = \mathbf{w}^\top\mathbf{x}$ where $\mathbf{w} = [\mathbf{w}_z, \mathbf{w}_y, \mathbf{w}_e]$ with default-ERM. Data from eq. (1) is always linearly separable due to the perfect stable feature, but many hyperplanes that separate the two classes exist. When a linear model is trained with default-ERM on linearly separable data, it achieves zero training loss and converges to the direction of a minimum $\ell_2$-norm solution that achieves a margin of at least 1 on all samples [Soudry et al., 2018, Wang et al., 2021, 2022]; this is called the max-margin solution. We now show that for a small enough leftover group, large enough scaling factor $B$ and dimension $d$ of the covariates, max-margin solutions depend more on the shortcut feature than the stable feature:

**Theorem 1.** *Let $\mathbf{w}^*$ be the max-margin predictor on $n$ training samples from eq. (1) with a leftover group of size $k$. There exist constants $C_1, C_2, N_0 > 0$ such that*

$$\forall\, n > N_0, \qquad \forall\ \text{integers } k \in \left(0, \frac{n}{10}\right), \qquad \forall\, d \geq C_1 k \log(3n), \qquad \forall\, B > C_2\sqrt{\frac{d}{k}}, \qquad (2)$$

*with probability at least $1 - 1/3n$ over draws of the training data, it holds that $B\mathbf{w}_z^* > \mathbf{w}_y^*$.*

The size of the leftover group $k$ concentrates around $(1 - \rho)n$ because each sample falls in the leftover group with probability $(1 - \rho)$. Thus, for $\rho > 0.9$, that is for a strong enough shortcut, the condition in theorem 1 that $k < n/10$ will hold with probability close to 1; see appendix A.5 for more details.

The proof is in appendix A. The first bit of intuition is that using the shortcut can have lower norm because of the scaling factor $B$. Using the shortcut only, however, misclassifies the leftover group. The next bit of intuition is that using noise from the leftover group increases margins in one group at a rate that scales with the dimension $d$, while the cost in the margin for the other group only grows as $\sqrt{d}$. This trade-off in margins means the leftover group can be correctly classified using noise without incorrectly classifying the shortcut group. The theorem then leverages convex duality to show that this type of classifier that uses the shortcut and noise has smaller $\ell_2$-norm than any linear classifier that uses the stable feature more.

The way the margin trade-off in the proof works is by constructing a linear classifier whose weights on the noise features are a scaled sum of the product of the label and the noise vector in the leftover group: for a scalar $\gamma$, the weights $\mathbf{w}_e = \gamma\sum_{i\in S_{\text{leftover}}} \mathbf{y}_i\boldsymbol{\delta}_i$. The margin change on the $j$th training sample from using these weights is $\mathbf{y}_j\mathbf{w}_e^\top\boldsymbol{\delta}_j$. For samples in the shortcut group, the margin change looks like a sum of mean zero independent and identically distributed variables; the standard deviation of this sum grows as $\sqrt{d}$. For samples in the leftover group, the margin change is the sum of mean one random variables; this sum grows as $d$ and its standard deviation grows as $\sqrt{d}$. The difference in mean relative to the standard deviation is what provides the trade-off in margins.

We now discuss three implications of the theorem.

**First, theorem 1 implies that the leftover group sees worse than random accuracy** (0.5). To see this, note that for samples in the leftover group the margin $\mathbf{y}(\mathbf{w}^*)^\top\mathbf{x} = \mathbf{w}_y^* - B\mathbf{w}_z^* + (\mathbf{w}_e^*)^\top\mathbf{y}\boldsymbol{\delta}$ is a Gaussian random variable centered at a negative number $\mathbf{w}_y^* - B\mathbf{w}_z^*$. Then, with $\Phi_e$ as the CDF of the zero-mean Gaussian random variable $(\mathbf{w}_e^*)^\top\boldsymbol{\delta}$, accuracy in the test leftover group is

$$p(\mathbf{y}(\mathbf{w}^*)^\top\mathbf{x} \geq 0 \mid \mathbf{y} \neq \mathbf{z}) = p[(\mathbf{w}_e^*)^\top\boldsymbol{\delta} > -(\mathbf{w}_y^* - B\mathbf{w}_z^*)] = 1 - \Phi_e(-(\mathbf{w}_y^* - B\mathbf{w}_z^*)) \leq 0.5.$$

Second, the leftover group in the training data is overfit in that the contribution of noise in prediction $(|(\mathbf{w}_e^*)^\top\boldsymbol{\delta}|)$ is greater than the contribution from the stable and shortcut features. Formally, in the training leftover group, $\mathbf{w}_y^* - B\mathbf{w}_z^* < 0$. Then, due to max-margin property,

$$\mathbf{w}_y^* - B\mathbf{w}_z^* + (\mathbf{w}_e^*)^\top\mathbf{y}_i\boldsymbol{\delta}_i > 1 \implies (\mathbf{w}_e^*)^\top\mathbf{y}_i\boldsymbol{\delta}_i \geq 1 - (\mathbf{w}_y^* - B\mathbf{w}_z^*) > |\mathbf{w}_y^* - B\mathbf{w}_z^*|.$$

Third, many works point to overparameterization as one of the causes behind shortcut learning [Sagawa et al., 2020a, Nagarajan et al., 2021, Wald et al., 2023], but in the setup in fig. 1, the

linear model has fewer parameters than samples in the training data. In such cases with non-overparameterized linear models, the choice of default-ERM is typically not questioned, especially when a feature exists that linearly separates the data. **Corollary 1 formally shows shortcut learning for non-overparameterized linear models. In words, default-ERM — that is vanilla logistic regression trained with gradient-based optimization — can yield models that rely more on the shortcut feature *even without overparameterization*.**

**Corollary 1.** *For all $n > N_0$ — where the constant $N_0$ is from [theorem 1](#) — with scalar $\tau \in (0, 1)$ such that the dimension of $\mathbf{x}$ is $d = \tau n < n$, for all integers $k < n \times \min \left\{ \frac{1}{10}, \frac{\tau}{C_1 \log 3n} \right\}$, a linear model trained via default-ERM yields a predictor $\mathbf{w}^*$ such that $B\mathbf{w}_z^* > \mathbf{w}_y^*$.*

If default-ERM produces models that suffer from shortcut learning even without overparameterization, its implicit inductive bias toward max-margin classification is inappropriate for perception tasks in the presence of shortcuts. Next, we study inductive biases more suited to perception tasks.

# 3   Toward inductive biases for perception tasks with shortcuts

The previous section formalized how default-ERM solutions, due to the max-margin inductive bias, rely on the shortcut and noise to minimize loss on training data even in the presence of a different zero-population-risk solution. Are there inductive biases more suitable for perception tasks?

Given a perfect stable feature $s(\mathbf{x})$ for a perception task, in that for a function $d$ when $\mathbf{y} = d \circ s(\mathbf{x})$, one can achieve margin $b \in (0, \infty)$ uniformly on all samples by predicting with the stable $b \cdot d \circ s(\mathbf{x})$. In contrast, max-margin classifiers allow for disparate margins as long as the smallest margin crosses 1, meaning that it does not impose uniform margins. The cost of allowing disparate margins is the preference for shortcuts even without overparamterization ([corollary 1](#)). In the same setting however, any uniform-margin classifier for the linear perception task ([eq. (1)](#)) relies only on the stable feature:

**Theorem 2.** *Consider $n$ samples of training data from DGP in [eq. (1)](#) with $d < n$. Consider a linear classifier $f_\theta(\mathbf{x}) = \mathbf{w}^\top \mathbf{x}$ such that for all samples in the training data $\mathbf{y}_i \mathbf{w}^\top \mathbf{x}_i = b$ for any $b \in (0, \infty)$. With probability 1 over draws of samples, $\mathbf{w} = [0, b, 0^{d-2}]$.*

[Theorem 2](#) shows that uniform-margin classifiers only depend on the stable feature, standing in contrast with max-margin classifiers which can depend on the shortcut feature ([theorem 1](#)). The proof is in [appendix A.6](#). **Thus, inductive biases toward uniform margins are better suited for perception tasks.** Next, we identify several ways to encourage uniform margins.

**Margin control (MARG-CTRL).**   To produce uniform margins with gradient-based optimization, we want the loss to be minimized at uniform-margin solutions and be gradient-optimizable. We identify a variety of losses that satisfy these properties, and we call them MARG-CTRL losses. MARG-CTRL losses have the property that per-sample loss monotonically decreases for margins until a threshold then increases for margins beyond it. In turn, minimizing loss then encourages all margins to move to the threshold.

Mechanically, when models depend more on shortcuts than the stable feature during training, margins on samples in the shortcut group will be larger than those in the leftover group; see the right panel in [fig. 1b](#) where the train loss in the shortcut group is lower than the leftover group indicating that the margins are smaller in the leftover group. This difference is margins is a consequence of the shortcut matching the label in one group and not the other, thus, encouraging the model to have similar margins across all samples pushes the model to depend less on the shortcut. In contrast, vanilla log-loss can be driven to zero in a direction with disparate margins across the groups as long as the margins on all samples go to $\infty$. We define MARG-CTRL losses for a model $f_\theta$ with the margin on a sample $(\mathbf{x}, \mathbf{y})$ defined as $\mathbf{y} f_\theta(\mathbf{x})$.

As the first MARG-CTRL loss, we develop the $\sigma$-damped log-loss: we evaluate log-loss on a margin multiplied by a monotonically decreasing function of the margin. In turn, the input to the loss increases with the margin till a point and then decreases. For a temperature $T$ and sigmoid function

$\sigma$, the $\sigma$-damped loss modifies the model output $f_\theta$ and plugs it into log-loss:

$$\ell_{\sigma\text{-damp}}(\mathbf{y}, f_\theta) = \ell_{\log}\left(\mathbf{y}\left(1 - \sigma\left(\frac{\mathbf{y}f_\theta}{T}\right)\right)f_\theta\right)$$

For large margin predictions $\mathbf{y}f_\theta > 0$, the term $1 - \sigma\left(\mathbf{y}f_\theta(\mathbf{x})/T\right)$ damps down the input to log-loss. The largest the input to $\ell_{\log}$ can get is $0.278T$, found by setting the derivative to zero, thus lower bounding the loss. As log-loss is a decreasing function of its input, the minimum of $\ell_{\sigma\text{-damp}}$ occurs when the margin is $0.278T$ on all samples. To demonstrate empirical advantage, we compare

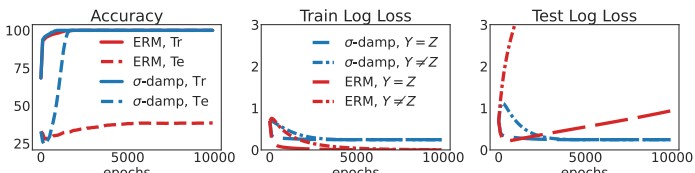

**Figure 2:** Using $\sigma$-damped log-loss yields linear models that depend on the perfect stable feature to achieve near perfect test accuracy. The middle panel shows that $\sigma$-damping maintains similar margins in the training shortcut and leftover groups unlike unconstrained log-loss, and the right panel shows $\sigma$-damp achieves better leftover test-loss.

standard log-loss to $\sigma$-damped loss on eq. (1); see fig. 2. The left panel of figure fig. 2 shows that test accuracy is better for $\sigma$-damp. The middle and right panels shows the effect of controlling margins in training, where losses on shortcut and leftover groups hover at the same value.

Second, we design the $\sigma$-stitch loss, which imitates log-loss when $\mathbf{y}f_\theta(\mathbf{x}) < u$ and penalizes larger margins ($\mathbf{y}f_\theta > u$) by negating the sign of $\mathbf{y}f_\theta(\mathbf{x})$:

$$\ell_{\sigma\text{-stitch}} = \ell_{log}\left(\mathbf{1}[\mathbf{y}f_\theta(\mathbf{x}) \leq u]\mathbf{y}f_\theta(\mathbf{x}) + \mathbf{1}[\mathbf{y}f_\theta(\mathbf{x}) > u](2u - \mathbf{y}f_\theta(\mathbf{x}))\right) \qquad (3)$$

As the third MARG-CTRL loss, we directly penalize large margins via a $\log$-penalty:

$$\ell_{\texttt{marg-log}} = \ell_{log}(\mathbf{y}f_\theta(\mathbf{x})) + \lambda \log\left(1 + |f_\theta(\mathbf{x})|^2\right) \qquad (4)$$

The fourth MARG-CTRL loss controls margins by penalizing $|f_\theta(\mathbf{x})|^2$:

$$\ell_{\text{SD}} = \ell_{log}(\mathbf{y}f_\theta(\mathbf{x})) + \lambda|f_\theta(\mathbf{x})|^2 \qquad (5)$$

This last penalty was called spectral decoupling (SD) by Pezeshki et al. [2021], who use it as a way to decouple learning dynamics in the neural tangent kernel (NTK) regime. Instead, from the lens of MARG-CTRL, SD mitigates shortcuts in eq. (1) because it encourages uniform margins, even though SD was originally derived from different principles, as we discuss in section 4. In appendix B.2, we plot all MARG-CTRL losses and show that MARG-CTRL improves over default-ERM on the linear perception task; see figs. 6 to 8. We also run MARG-CTRL on a neural network and show that while default-ERM achieves test accuracy worse than random chance, MARG-CTRL achieves $100\%$ test accuracy; see figs. 10 to 13 in appendix B.4.

## 4 Related work

A large body of work tackles shortcut learning under different assumptions [Arjovsky et al., 2019, Wald et al., 2021, Krueger et al., 2020, Creager et al., 2021, Veitch et al., 2021, Puli et al., 2022, Heinze-Deml and Meinshausen, 2021, Belinkov and Bisk, 2017]. A different line of work focuses on learning in neural networks in idealized settings [Yang and Salman, 2019, Ronen et al., 2019, Jo and Bengio, 2017, Baker et al., 2018, Saxe et al., 2013, Gidel et al., 2019, Advani et al., 2020].

Shah et al. [2020] study simplicity bias [Valle-Perez et al., 2018] and show that neural networks provably learn the linear function over a non-linear one, in the first epoch of training. In a similar vein, Hermann and Lampinen [2020] show that neural networks can prefer a linearly-decodable feature over a non-linear but more predictive feature, and Scimeca et al. [2021] make similar observations and use loss landscapes to empirically study which features are easier to learn. Simplicity bias alone only describes neural biases early in training and does not explain why more predictive stable features are not learned later. Unlike simplicity bias which focuses on linear versus non-linear features, max-margin bias is the reason default-ERM prefers one linear feature, the shortcut, over another, the stable feature, like in the synthetic experiment in section 2.

While Pezeshki et al. [2021] allow for perfect features, they hypothesize that shortcut learning occurs because when one feature is learned first, other features are gradient-starved and are not learned as

well. They focus on a special setting where feature representations for different samples have inner product equal to a small constant to show that models can depend more on the imperfect feature than the perfect feature. In this special setting, they show that penalizing the magnitudes of what we call the margin mitigates shortcuts; this method is called spectral decoupling (SD). However, as we show in appendix B.5, the assumption in Lemma 1 [Pezeshki et al., 2021] is violated when using a linear model to classify in the simple linear DGP in eq. (1). However, SD on a linear model mitigates shortcuts in the DGP in eq. (1); see B.5. Thus, the theory in Pezeshki et al. [2021] fails to not explain why SD works for eq. (1), but the uniform-margin property explains why all the MARG-CTRL losses, including SD, mitigate shortcuts.

Nagarajan et al. [2021] consider tasks with perfect stable features and formalize geometric properties of the data that make max-margin classifiers have non-zero weight on the shortcut feature ($\mathbf{w}_z > 0$). In their set up, the linear models are overparameterized and it is unclear when $\mathbf{w}_z > 0$ leads to worse-than-random accuracy in the leftover group because they do not separate the model's dependence on the stable feature from the dependence on noise. See fig. 14 for an example where $\mathbf{w}_z > 0$ but test accuracy is $100\%$. In contrast to Nagarajan et al. [2021], theorem 1 gives DGPs where the leftover group accuracy is worse than random, even without overparameterization. Ahuja et al. [2021] also consider linear classification with default-ERM with a perfect stable feature and conclude that default-ERM learns only the stable feature because they assume no additional dimensions of noise in the covariates. We develop the necessary nuance by including noise in the problem and showing default-ERM depends on the shortcut feature even without overparameterization.

Sagawa et al. [2020b] and Wald et al. [2023] both consider overparameterized settings where the shortcut feature is informative of the label even after conditioning on the stable feature. In both cases, the Bayes-optimal predictor also depends on the shortcut feature, which means their settings do not allow for an explanation of shortcut dependence in examples like fig. 1. In contrast, we show shortcut dependence occurs even in the presence of a perfect stable feature and without overparameterization. Li et al. [2019], Pezeshki et al. [2022] focus on relative feature complexity and discuss the effects of large learning rate (LR) on which features are learned first during training, but do not allow for perfect features. Idrissi et al. [2022] empirically find that tuning LR and weight decay (WD) gets default-ERM to perform similar to two-stage shortcut-mitigating methods like Just Train Twice (JTT) [Liu et al., 2021]. We view the findings of [Idrissi et al., 2022] through the lens of MARG-CTRL and explain how large LR and WD approximate MARG-CTRL to mitigate shortcuts; see section 5.

MARG-CTRL is related to but different from methods proposed in Liu et al. [2017], Cao et al. [2019], Kini et al. [2021]. These works normalize representations or the last linear layers and linearly transform the logits to learn models with better margins under label imbalance. Next, methods like Learning from Failure (LFF) [Nam et al., 2020], JTT [Liu et al., 2021], and Correct-n-Contrast (CNC) [Zhang et al., 2022] build two-stage procedures to avoid shortcut learning without group annotations in training. They assume that default-ERM produces models that depend more on the shortcut and select hyperparamters of the two stage process using validation group annotations. In the nuisance-free setting where there are no validation group annotations, the performance of these methods can degrade below that of default-ERM. In contrast, better characterizing the source of shortcut learning in perceptual problems leads to MARG-CTRL methods that are not as reliant on validation group annotations (see nuisance-free results in Section 5). **Without any group annotations, encouraging uniform margins via MARG-CTRL mitigates shortcuts better than JTT and CNC.**

Soudry et al. [2018] characterize the inductive bias of gradient descent to converge in direction to max-margin solutions when using exponentially tailed loses; Wang et al. [2021, 2022] then prove similar biases toward max-margin solutions for Adam and RMSProp. Ji et al. [2020] show that for general losses that decrease in $\mathbf{y} f_\theta(\mathbf{x})$, gradient descent has an inductive bias to follow the $\ell_2$-regularization path. All these inductive biases prefer shortcuts if using them leads to lower loss within an $\ell_2$-norm-budget. MARG-CTRL provides a different inductive bias toward producing the same margin on all samples, which means gradient descent veers models away from imperfect shortcuts that lead to disparity in network outputs. Such inductive biases are suitable for tasks where a feature determines the label ($h(\mathbf{x}) = \mathbf{y}$).

## 5 Vision and language experiments

We evaluate MARG-CTRL on common datasets with shortcuts: Waterbirds, CelebA [Sagawa et al., 2020a], and Civilcomments [Koh et al., 2021]. First, MARG-CTRL always improves over default-ERM. Then, we show that MARG-CTRL performs similar to or better than two-stage shortcut-

mitigating methods like JTT [Liu et al., 2021] and CNC [Zhang et al., 2022] in traditional evaluation settings where group annotations are available in the validation data. Finally, we introduce a more challenging setting that only provides class labels in training and validation, called the **nuisance-free setting**. In contrast to the traditional setting that always assumes validation group annotations, the nuisance-free setting does not provide group annotations in either training or in validation. In the nuisance-free setting, MARG-CTRL outperforms JTT and CNC, even though the latter are supposed to mitigate shortcuts without knowledge of the groups.

**Datasets.** We use the Waterbirds and CelebA datasets from Sagawa et al. [2020a] and the Civil-Comments dataset from Borkan et al. [2019], Koh et al. [2021]. In Waterbirds, the task is to classify images of a waterbird or landbird, and the label is spuriously correlated with the image background consisting of land or water. There are two types of birds and two types of background, leading to a total of 4 groups defined by values of $y, z$. In CelebA [Liu et al., 2015, Sagawa et al., 2020a], the task is to classify hair color of celebrities as blond or not. The gender of the celebrity is a shortcut for hair color. There are two types of hair color and two genders in this dataset, leading to a total of 4 groups defined by values of $y, z$. In CivilComments-WILDS [Borkan et al., 2019, Koh et al., 2021], the task is to classify whether an online comment is toxic or non-toxic, and the label is spuriously correlated with mentions of certain demographic identities. There are 2 labels and 8 types of the shortcut features, leading to 16 groups.

**Metrics, model selection, and hyperparameters.** We report the worst-group test accuracy for each method. The groups are defined based on the labels and shortcut features. The more a model depends on the shortcut, the worse the worst-group error. Due to the label imbalance in all the datasets, we use variants of $\sigma$-damp, $\sigma$-stitch, MARG-LOG, and SD with class-dependent hyperparameters; see appendix B.6.2. For all methods, we use the standard Adam optimizer [Kingma and Ba, 2015] and let the learning rate and weight decay hyperparameters be tuned along with the method's hyperparameters. We first report results for all methods using validation worst-group accuracy to select method and optimization hyperparameters and early stop. For both JTT and CNC, this is the evaluation setting that is used in

|        | CelebA         | WB             | Civil          |
|--------|----------------|----------------|----------------|
| ERM    | $72.8 \pm 9.4$ | $70.8 \pm 2.4$ | $60.1 \pm 0.4$ |
| CNC    | $81.1 \pm 0.6$ | $68.0 \pm 1.8$ | $68.8 \pm 0.2$ |
| JTT    | $75.2 \pm 4.6$ | $71.7 \pm 4.0$ | $69.9 \pm 0.4$ |
| marg-log | $82.8 \pm 1.1$ | $78.2 \pm 1.9$ | $68.4 \pm 1.8$ |
| $\sigma$-damp | $79.4 \pm 0.6$ | $78.6 \pm 1.1$ | $69.6 \pm 0.4$ |
| SD     | $81.4 \pm 2.5$ | $80.5 \pm 1.4$ | $69.9 \pm 1.1$ |
| $\sigma$-stitch | $81.1 \pm 2.2$ | $75.9 \pm 3.4$ | $67.8 \pm 2.8$ |

**Table 1:** Mean and standard deviation of test worst-group accuracies over two seeds for default-ERM, JTT, CNC, $\sigma$-damp, $\sigma$-stitch, SD, and `marg-log`. Every MARG-CTRL method outperforms default-ERM on every dataset. On Waterbirds, MARG-CTRL outperforms JTT and CNC. On CelebA, SD, `marg-log`, and $\sigma$-stitch beat JTT and achieve similar or better performance than CNC. On Civil-Comments, $\sigma$-damp and SD beat CNC and achieve similar performance to JTT.

existing work [Liu et al., 2021, Idrissi et al., 2022, Zhang et al., 2022]. Finally, in the nuisance-free setting where no group annotations are available, we select hyperparameters using label-balanced average accuracy. Appendix B.6 gives further details about the training, hyperparameters, and experimental results.

## 5.1 MARG-CTRL mitigates shortcuts in the default setting

Here, we experiment in the standard setting from Liu et al. [2021], Idrissi et al. [2022], Zhang et al. [2022] and use validation group annotations to tune hyperparameters and early-stopping.

**MARG-CTRL improves over default-ERM.** We compare MARG-CTRL to default-ERM on CelebA, Waterbirds, and Civilcomments. Table 1 shows that every MARG-CTRL method achieves higher test worst-group accuracy than default-ERM on all datasets. Default-ERM achieves a mean test worst-group accuracy of $70.8\%, 72.8\%$ and $60.1\%$ on Waterbirds, CelebA, and Civilcomments respectively. Compared to default-ERM, MARG-CTRL methods provide a $5 - 10\%$ improvement on Waterbirds, $7 - 10\%$ improvement on CelebA, $7 - 10\%$ improvement on Civilcomments. These improvements show the value of inductive biases more suitable for perception tasks.

**Large LR and WD may imitate MARG-CTRL in ERM.** Default-ERM's performance varies greatly across different values of LR and WD on, for instance, CelebA: the test worst-group accuracy improves by more than 20 points over different LR and WD combinations. Why does tuning LR and WD yield such improvements? We explain this phenomenon as a consequence of instability in optimiza-

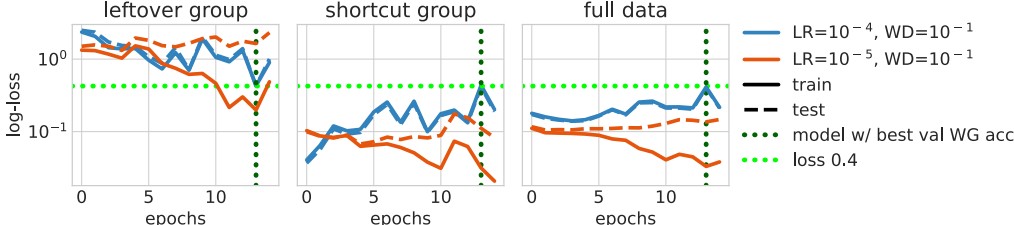

**Figure 3:** Loss curves of default-ERM on CelebA for two combinations of LR and WD. The combination with the larger learning rate (blue) achieves 72.8% test worst-group accuracy, beating the other combination by 20%. The model that achieves the best validation (and test) worst-group accuracy is the one at epoch 13 from the blue run. This model achieves similar loss in both groups and the full data model suggesting that large LR and WD controls margins from exploding (higher training loss in all panels) and avoids systematically smaller margins in the leftover group compared to the shortcut group.

tion induced by large LR and WD which prevents the model from maximizing margins and in turn can control margins. Figure 3 provides evidence for this explanation by comparing default-ERM's loss curves for two LR and WD combinations.

The blue curves in fig. 3 correspond to the run with the larger LR and WD combination. The model that achieves the best validation (and test) worst-group accuracy over all combinations of hyperparameters for default-ERM, including those not in the plot, is the one at epoch 13 on the blue curves. This model achieves similar train and test losses ($\approx 0.4$) and thus similar margins in the shortcut group, the leftover group, and the whole dataset. The red curves stand in contrast where the lower LR results in the leftover group having higher training and test losses, and therefore smaller margins, compared to the shortcut group. These observations together support the explanation that default-ERM with large LR and WD mitigates shortcuts when controlling margins like MARG-CTRL.

**MARG-CTRL performs as well or better than two-stage shortcut-mitigating methods.** Two-stage shortcut mitigating methods like Correct-n-Contrast (CNC) and Just Train Twice (JTT) aim to mitigate shortcuts by using a model trained with default-ERM to approximate group annotations. They rely on the assumption that a model trained via default-ERM either predicts with the shortcut feature (like background in Waterbirds) or that the model's representations separate into clusters based on the shortcut feature. The methods then approximate group annotations using this default-ERM-trained model and use them to mitigate shortcut learning in a second predictive model. JTT upweights the loss on the approximate leftover group and CNC uses a contrastive loss to enforce the model's representations to be similar across samples that have the same label but different approximate group annotations. Appendix B.6.1 gives details.

Table 1 compares MARG-CTRL to JTT and CNC on Waterbirds, Celeba, and CivilComments. On CelebA, SD, `marg-log`, and $\sigma$-stitch perform similar to CNC while all MARG-CTRL techniques outperform JTT. On Waterbirds, all MARG-CTRL methods outperform JTT and CNC. On CivilComments, $\sigma$-damp and SD perform similar to JTT and outperform CNC. CNC's performance on Waterbirds differs from Zhang et al. [2022] because their reported performance requires unique large WD choices (like WD set to 1) to build a first-stage model that relies most on the shortcut feature without overfitting to the training data.

**MARG-CTRL is faster than JTT and CNC.** MARG-CTRL takes the same time as default-ERM, taking around $1, 20$ and $60$ minutes per epoch for Waterbirds, CelebA, and CivilComments respectively on an RTX8000 GPU. In contrast, on average over runs, JTT takes around $6, 80, 120$ minutes per epoch and CNC takes around $8, 180, 360$ minutes per epoch. Thus, MARG-CTRL performs as well or better than JTT and CNC while being simpler to implement and computationally cheaper.

## 5.2 MARG-CTRL mitigates shortcuts in the nuisance-free setting

Work like [Liu et al., 2021, Zhang et al., 2022] crucially require validation group annotations because these methods push the work of selecting models for mitigating shortcuts to validation. Determining shortcuts itself is a laborious manual process, which means group annotations will often be unavailable. Further, given a perfect stable feature that determines the label and a shortcut that does not, only models that rely on the stable feature more than the shortcut achieve the highest validation ac-

curacy. Thus, we introduce a more challenging setting that only provides class labels in training and validation, called the **nuisance-free setting**. In the nuisance-free setting, models are selected based on label-balanced average accuracy: the average of the accuracies over samples of each class.

Table 2 reports test worst-group (WG) accuracy in the nuisance-free setting. **On all the datasets, every MARG-CTRL outperforms default-ERM, JTT, and CNC.** On average, the MARG-CTRL methods close at least $61\%$ of the gap between default-ERM in the nuisance-free setting and the best performance in table 1 on every dataset. **In contrast, CNC and JTT sometimes perform worse than default-ERM.**

|  | CelebA | WB | Civil |
|---|---|---|---|
| ERM | $57.5 \pm 5.8$ | $69.1 \pm 2.1$ | $60.7 \pm 1.5$ |
| CNC | $67.8 \pm 0.6$ | $60.0 \pm 8.0$ | $61.4 \pm 1.9$ |
| JTT | $53.3 \pm 3.3$ | $71.7 \pm 4.0$ | $53.4 \pm 2.1$ |
| marg-log | $74.2 \pm 1.4$ | $77.9 \pm 0.3$ | $66.8 \pm 0.2$ |
| $\sigma$-damp | $70.8 \pm 0.3$ | $74.8 \pm 1.6$ | $65.6 \pm 0.2$ |
| SD | $70.3 \pm 0.3$ | $78.7 \pm 1.4$ | $67.8 \pm 1.3$ |
| $\sigma$-stitch | $76.7 \pm 0.6$ | $74.5 \pm 1.2$ | $66.0 \pm 1.0$ |

**Table 2:** Average and standard deviation of test worst-group accuracy over two seeds of MARG-CTRL, default-ERM, JTT, and CNC in the nuisance-free setting. Hyperparameter selection and early stopping use label-balanced average accuracy. All MARG-CTRL methods outperform default-ERM, JTT, and CNC on all datasets.

## 6 Discussion

We study why default-ERM — gradient-based optimization of log-loss — yields models that depend on the shortcut even when the population minimum of log-loss is achieved by models that depend only on the stable feature. By studying a linear task with perfect stable features, we show that default-ERM's preference toward shortcuts sprouts from an inductive bias toward maximizing margins. Instead, inductive biases toward uniform margins improve dependence on the stable feature and can be implemented via MARG-CTRL. MARG-CTRL improves over default-ERM on a variety of perception tasks in vision and language without group annotations in training, and is competitive with more expensive two-stage shortcut-mitigating methods. In the nuisance-free setting, where even validation group annotations are unavailable, MARG-CTRL outperforms all the baselines. The performance that MARG-CTRL yields demonstrates that changing inductive biases can remove the need for expensive shortcut-mitigating methods in perception tasks.

Without overparameterization, uniform-margin classifiers are unique and learn stable features only, while max-margin classifiers can depend more on shortcuts. With overparameterization, max-margin classifiers are still unique but uniform-margin solutions are not which necessitates choosing between solutions. The experiments in section 5 suggest that choosing between uniform-margin classifiers with penalties like $\ell_2$ improves over max-margin classifiers with $\ell_2$: all experiments use overparameterized models trained with weight decay and MARG-CTRL outperforms default-ERM. Further, our experiments suggest that uniform-margin classifiers are insensitive to the WD and LR choices, unlike max-margin classifiers; appendix B.8 shows that MARG-CTRL achieves high performance for all LR and WD choices but ERM requires tuning.

Theorem 1 also explains how balancing may or may not improve dependence on the stable features. For example, a weighting-based approach produces the same max-margin solution as default-ERM [Sagawa et al., 2020b, Rosset et al., 2003], but subsampling leads to a different solution that could depend less on the shortcut. For the latter however, models are more prone to overfitting on the smaller subsampled dataset. Similar observations were made in Sagawa et al. [2020b] but this work extends the insight to tasks with perfect stable features. Comparing ERM and MARG-CTRL on subsampled data would be fruitful.

Any exponentially tailed loss when minimized via gradient descent converges to the max-margin solution in direction [Soudry et al., 2018]. Thus, theorem 1 characterizes shortcut learning for any exponentially-tailed loss. However, losses with decreasing polynomial tails — for example, $\ell(a) = \frac{1}{1+a^K}$ for some $K > 0$ — do not converge to the max-margin classifier. One future direction is to show shortcut-dependence results like theorem 1 for polynomial-tailed losses, which in turn would mean that all common classification losses with a decreasing tail impose inductive biases unsuitable for perception tasks.

In the tasks we consider with perfect stable features, Bayes-optimal predictors rely only on the stable feature. A weaker independence condition implies the same property of Bayes-optimal predictors even when $\mathbf{y}$ is not determined by $s(\mathbf{x})$: $\mathbf{y} \perp\!\!\!\perp (\mathbf{x}, \mathbf{z}) \mid s(\mathbf{x})$. For example, in the CivilComments dataset a few instances have ambiguous labels [Xenos et al., 2022] meaning that there may not be a perfect stable feature. Studying uniform margins and other inductive biases under this independence would be fruitful.

**Acknowledgements.** This work was supported by the NIH/NHLBI Award R01HL148248, NSF Award 1922658 NRT-HDR: FUTURE Foundations, Translation, and Responsibility for Data Science, NSF CAREER Award 2145542, ONR N00014-23-1-2634, Google. Aahlad Puli is partly supported by the Apple Scholars in AI/ML PhD fellowship.

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

# A    Appendix: Proof of Theorem 1, Corollary 1, and Theorem 2

## A.1    Helper Lemmas

### A.1.1    Bounding norms and inner products of isotropic random vectors.

The main lemmas of this section are lemma 3 and lemma 4. We will then use these two to bound norms of sums of random vectors and inner products between the sum and a single random vector in lemma 5. We first list some facts from [Vershynin, 2018] that we will use to bound the probability with which norms and inner products of Gaussian random vectors deviate far from their mean.

**Definition 1.** *(Sub-Gaussian norm) For an r.v.* $\mathbf{x}$, *the sub-Gaussian norm, or* $\psi_2$-*norm, is*

$$\|\mathbf{x}\|_{\psi_2} = \inf\{t > 0, \mathbb{E}[\exp(\mathbf{x}^2/t^2)] \leq 2\}.$$

*An r.v. is called sub-Gaussian if its* $\psi_2$-*norm is finite and for some fixed constant* $c$

$$p(|\mathbf{x}| > t) \leq 2\exp(-ct^2/\|\mathbf{x}\|_{\psi_2}).$$

*A Gaussian r.v.* $\mathbf{x} \sim \mathcal{N}(0, \sigma^2)$ *has an* $\psi_2$-*norm of* $G\sigma$ *for a constant* $G = \sqrt{\frac{8}{3}}$.[2]

**Definition 2.** *(Sub-exponential norm) For an r.v.* $\mathbf{x}$, *the sub-exponential norm, or* $\psi_1$-*norm, is*

$$\|\mathbf{x}\|_{\psi_1} = \inf\{t > 0, \mathbb{E}[\exp(|\mathbf{x}|/t)] \leq 2\}.$$

*A sub-exponential r.v. is one that has finite* $\psi_1$-*norm.*

**Lemma 1.** *(Lemma 2.7.7 from [Vershynin, 2018]) Products of sub-Gaussian random variables* $\mathbf{x}, \mathbf{y}$ *is a sub-exponential random variable with it's* $\psi_1$-*norm bounded by the product of the* $\psi_2$-*norm*

$$\|\mathbf{x}\mathbf{y}\|_{\psi_1} \leq \|\mathbf{x}\|_{\psi_2}\|\mathbf{y}\|_{\psi_2}$$

Lemma 1 implies that the product of two mean-zero standard normal vectors is a sub-exponential random variable with $\psi_1$-norm less than $G^2$.

**Lemma 2.** *(Bernstein inequality, Theorem 2.8.2 [Vershynin, 2018]) For i.i.d sub-exponential random variables* $\mathbf{x}_1, \cdots, \mathbf{x}_d$, *for a fixed constant* $c = \frac{1}{(2e)^2}$ *and* $K = \|\mathbf{x}_1\|_{\psi_1}$

$$p\left(\left|\sum_{i=1}^{d} \mathbf{x}_i\right| > t\right) \leq 2\exp\left(-c\min\left\{\frac{t^2}{K^2 d}, \frac{t}{K}\right\}\right)$$

Next, we apply these facts to bound the sizes of inner products between two unit-variance Gaussian vectors.

**Lemma 3.** *(Bounds on inner products of Gaussian vectors) Let* $\mathbf{u}, \mathbf{v}$ *be* $d$-*dimensional random vectors where each coordinate is an i.i.d standard normal r.v. Then, for any scalar* $\epsilon > 0$ *such that* $\epsilon \leq G^2\sqrt{d}$, *for a fixed constant* $c = \frac{1}{(2e)^2}$

$$p\left(\left|\mathbf{u}^\top\mathbf{v}\right| > \epsilon\sqrt{d}\right) \leq 2\exp\left(-c\frac{\epsilon^2}{G^4}\right).$$

---

[2] $G = \sqrt{\frac{8}{3}}$. This follows from:

$$\mathbb{E}_{\mathbf{x}\sim\mathcal{N}(0,\sigma^2)}[\exp(\mathbf{x}^2/t^2)] = \int_{-\infty}^{\infty}\frac{1}{\sigma\sqrt{2\pi}}\exp(-x^2/2\sigma^2)\exp(x^2/t^2)dx = \int_{-\infty}^{\infty}\frac{1}{\sigma\sqrt{2\pi}}\exp\left(-x^2\frac{(t^2-2\sigma^2)}{2\sigma^2 t^2}\right)dx$$

$$= \frac{1}{\sigma\sqrt{2\pi}}\sqrt{\frac{\pi}{\frac{(t^2-2\sigma^2)}{2\sigma^2 t^2}}} = \frac{1}{\sigma\sqrt{\pi}}\sqrt{\frac{\pi\sigma^2 t^2}{(t^2-2\sigma^2)}} = \sqrt{\frac{t^2}{(t^2-2\sigma^2)}}$$

$$\sqrt{\frac{t^2}{(t^2-2\sigma^2)}} \leq 2 \implies t^2 \leq 4(t^2-2\sigma^2) \implies 8\sigma^2 \leq 3t^2 \implies \inf\{t : 8\sigma^2 \leq 3t^2\} = \sqrt{\frac{8}{3}}\sigma.$$

*Proof.* First, the inner product is $\mathbf{u}^\top \mathbf{v} = \sum_i^d \mathbf{u}_i \mathbf{v}_i$; it is the sum of products of i.i.d. standard normal r.v. ($\sigma = 1$). Then, by lemma 1, each term in the sum is a sub-exponential r.v. with $\psi_1$-norm bounded as follows:

$$K = \|\mathbf{u}_i \mathbf{v}_i\|_{\psi_1} \le \|\mathbf{u}_i\|_{\psi_2} \|\mathbf{u}_i\|_{\psi_2} = G \times G = G^2. \tag{6}$$

We can apply Bernstein inequality lemma 2 to sub-exponential r.v. to the inner product and then upper bound the probability by replacing $K$ with the larger $G^2$ in eq. (6)

$$p\left(|\mathbf{u}^\top \mathbf{v}| > t\right) \le 2\exp\left(-c\min\left\{\frac{t^2}{K^2 d}, \frac{t}{K}\right\}\right) \le 2\exp\left(-c\min\left\{\frac{t^2}{G^4 d}, \frac{t}{G^2}\right\}\right)$$

Substituting $t = \epsilon\sqrt{d}$ in the above gives us:

$$p\left(|\mathbf{u}^\top \mathbf{v}| > \epsilon\sqrt{d}\right) \le 2\exp\left(-c\min\left\{\frac{\epsilon^2 d}{G^4 d}, \frac{\epsilon\sqrt{d}}{G^2}\right\}\right)$$

Using the fact that $\epsilon \le G^2\sqrt{d}$ to achieve the minimum concludes the proof:

$$\epsilon \le G^2\sqrt{d} \implies \epsilon^2 \le \epsilon G^2\sqrt{d} \implies \frac{\epsilon^2}{G^4} \le \frac{\epsilon\sqrt{d}}{G^2} \implies \min\left\{\frac{\epsilon^2 d}{G^4 d}, \frac{\epsilon\sqrt{d}}{G^2}\right\} = \frac{\epsilon^2}{G^4}$$

$\square$

**Lemma 4.** *Let $\mathbf{x}$ be a Gaussian vector of size $d$ where each element is a standard normal, meaning that $\|\mathbf{x}_i\|_{\psi_2} = G$. Then, for any $t > 0$ and a fixed constant $c = \frac{1}{(2e)^2}$, the norm of the vector concentrates around $\sqrt{d}$ according to*

$$p\left(\left|\|\mathbf{x}\| - \sqrt{d}\right| > t\right) \le 2\exp(-ct^2/G^4).$$

*Proof.* Equation 3.3 from the proof of theorem 3.1.1 in [Vershynin, 2018] shows that

$$p(|\|\mathbf{x}\| - \sqrt{d}| > t) \le 2\exp(-ct^2/(\max_i \|\mathbf{x}_i\|_{\psi_2})^4).$$

As $\mathbf{x}$ has i.i.d standard normal entries, $\max_i \|\mathbf{x}_i\|_{\psi_2} = G$, concluding the proof. $\square$

### A.1.2 Concentration of norms of sums of random vectors and their inner products

This is the main lemma that we will use in proving theorem 1.

**Lemma 5.** *Consider a set of vectors $V = \{\boldsymbol{\delta}_i\}$ where $\boldsymbol{\delta}_i \in \mathbf{R}^d$ of size $T_V \ge 1$ where each element of each vector is drawn independently from the standard normal distribution $\mathcal{N}(0, 1)$. Then, for a fixed constant $c = \frac{1}{(2e)^2}$ and any $\epsilon \in (0, G^2\sqrt{d})$ with probability $\ge 1 - 2\exp(-\epsilon^2 \frac{c}{G^4})$*

$$\left\|\frac{1}{\sqrt{T_V}}\sum_{i\in V}\boldsymbol{\delta}_i\right\| \le \sqrt{d} + \epsilon \tag{7}$$

*and with probability $\ge 1 - 4T_V \exp(-\epsilon^2 \frac{c}{G^4})$*

$$\forall \boldsymbol{\delta}_j \in V \quad \left\langle \boldsymbol{\delta}_j, \sum_{i\in V}\boldsymbol{\delta}_i\right\rangle \ge d - 3\epsilon\sqrt{T_V d} \tag{8}$$

*Further, consider any set $U$ of vectors $U = \{\boldsymbol{\delta}_i\}$ of size $T_U$, where each vector also has coordinates drawn i.i.d from the standard normal distribution $\mathcal{N}(0, 1)$. Then, with probability $\ge 1 - 2T_u \exp(-\epsilon^2 \frac{c}{G^4})$*

$$\forall \boldsymbol{\delta}_j \in U \quad \left|\left\langle \boldsymbol{\delta}_j, \sum_{i\in V}\boldsymbol{\delta}_i\right\rangle\right| \le \epsilon\sqrt{T_V d}, \tag{9}$$

*By union bound, the three events above hold at once with a probability at least $1 - 2(2T_V + T_u + 1)\exp(-\epsilon^2 \frac{c}{G^4})$.*

*Proof.* We split the proof into three parts one each for eqs. (7) to (9).

**Proof of eq. (7).** As $\boldsymbol{\delta}$ is a vector of random i.i.d standard normal random variables, note that $\frac{1}{\sqrt{T_V}}\sum_i \boldsymbol{\delta}_i$ is also a vector of i.i.d standard normal random variables. This follows from the fact that the sum of $T_V$ standard normal random variables is a mean-zero Gaussian random variable with standard deviation $\sqrt{T_V}$. Thus dividing by the standard deviation makes the variance 1, making it standard normal.

Then, applying lemma 4 with $t = \epsilon$ gives us the following bound:

$$p\left(\left\|\frac{1}{\sqrt{T_V}}\sum_i \boldsymbol{\delta}_i\right\| > \sqrt{d} + \epsilon\right) \leq p\left(\left|\left\|\frac{1}{\sqrt{T_V}}\sum_i \boldsymbol{\delta}_i\right\| - \sqrt{d}\right| > \epsilon\right) \leq 2\exp(-c\epsilon^2/G^4)$$

**Proof of eq. (8)** We split the inner product into two cases: $T_V = 1$ and $T_V \geq 2$.

**Case $T_V = 1$.** First note that due to lemma 4,

$$\forall j \in V, \qquad p\left(\|\boldsymbol{\delta}_j\| < \sqrt{d} - \epsilon\right) \leq p\left(\left|\|\boldsymbol{\delta}_j\| - \sqrt{d}\right| > \epsilon\right) \leq 2\exp(-c\epsilon^2/G^4).$$

Then, the following lower bound holds with probability at least $1 - 2\exp(-c\epsilon^2/G^4)$

$$\forall j \in V, \qquad \left\langle \boldsymbol{\delta}_j, \sum_{i \in V} \boldsymbol{\delta}_i \right\rangle = \|\boldsymbol{\delta}_j\|^2$$
$$\geq (\sqrt{d} - \epsilon)^2$$
$$\geq d - 2\epsilon\sqrt{d}$$
$$\geq d - 3\epsilon\sqrt{T_V d},$$

To summarize this case, with the fact that $1 - 2\exp(-c\epsilon^2/G^4) \geq 1 - 4T_V\exp(-c\epsilon^2/G^4)$, we have that

$$\forall j \in V, \qquad \left\langle \boldsymbol{\delta}_j, \sum_{i \in V} \boldsymbol{\delta}_i \right\rangle \geq d - 3\epsilon\sqrt{T_V d},$$

with probability at least $1 - 4T_V\exp(-c\epsilon^2/G^4)$.

**Case $T_V \geq 2$.** First note that,

$$\forall j \in V \qquad \left\langle \boldsymbol{\delta}_j, \sum_{i \in V} \boldsymbol{\delta}_i \right\rangle = \|\boldsymbol{\delta}_j\|^2 + \left\langle \boldsymbol{\delta}_j, \sum_{i \in V, i \neq j} \boldsymbol{\delta}_i \right\rangle$$

For each of the $T_V$ different $\boldsymbol{\delta}_j$'s, using lemma 4 bounds the probability of the norm $\|\boldsymbol{\delta}_j\|$ being larger than $\sqrt{d} - \epsilon$:

$$p\left(\|\boldsymbol{\delta}_j\| < \sqrt{d} - \epsilon\right) \leq p\left(\left|\|\boldsymbol{\delta}_j\| - \sqrt{d}\right| > \epsilon\right) \leq 2\exp(-c\epsilon^2/G^4).$$

In the case where $T_V \geq 2$, we express the inner product of a vector and a sum of vectors as follows

$$\left\langle \boldsymbol{\delta}_j, \sum_{i \in V, i \neq j} \boldsymbol{\delta}_i \right\rangle = \sqrt{T_V - 1}\left\langle \boldsymbol{\delta}_j, \frac{1}{\sqrt{T_V - 1}}\sum_{i \in V, i \neq j} \boldsymbol{\delta}_i \right\rangle,$$

and noting that like above, $\frac{1}{\sqrt{T_V - 1}}\sum_{i \in V, i \neq j} \boldsymbol{\delta}_i$ is a vector of standard normal random variables, we apply lemma 3 to get

$$\forall i \in V \qquad p\left(\left|\left\langle \boldsymbol{\delta}_j, \sum_{i \in V, i \neq j} \boldsymbol{\delta}_i \right\rangle\right| \geq \epsilon\sqrt{(T_V - 1)d}\right) \leq 2\exp\left(-c\frac{\epsilon^2}{G^4}\right).$$

Putting these together, by union bound over $V$

$$p\left[\forall j \in V \qquad \left(\|\boldsymbol{\delta}_j\| < \sqrt{d} - \epsilon\right) \text{ or } \left(\left|\left\langle \boldsymbol{\delta}_j, \sum_{i \in V, i \neq j} \boldsymbol{\delta}_i \right\rangle\right| \geq \epsilon\sqrt{(T_V - 1)d}\right)\right]$$

$$\leq \sum_{j \in V} p\left(\|\boldsymbol{\delta}_j\| < \sqrt{d} - \epsilon\right) + p\left(\left|\left\langle \boldsymbol{\delta}_j, \sum_{i \in V, i \neq j} \boldsymbol{\delta}_i \right\rangle\right| \geq \epsilon\sqrt{(T_V - 1)d}\right)$$

$$\leq \sum_{j \in V} 2\exp\left(-c\frac{\epsilon^2}{G^4}\right) + 2\exp\left(-c\frac{\epsilon^2}{G^4}\right)$$

$$\leq 4T_V \exp\left(-c\frac{\epsilon^2}{G^4}\right).$$

Thus, with probability at least $1 - 4T_V \exp\left(-c\frac{\epsilon^2}{G^4}\right)$, none of the events happen and

$$\forall j \in V \qquad \left\langle \boldsymbol{\delta}_j, \sum_{i \in V} \boldsymbol{\delta}_i \right\rangle = \|\boldsymbol{\delta}_j\|^2 + \left\langle \boldsymbol{\delta}_j, \sum_{i \in V, i \neq j} \boldsymbol{\delta}_i \right\rangle$$

$$\geq (\sqrt{d} - \epsilon)^2 - \epsilon\sqrt{(T_V - 1)d}$$

$$= d - 2\epsilon\sqrt{d} + \epsilon^2 - \epsilon\sqrt{(T_V - 1)d}$$

$$\geq d - 2\epsilon\sqrt{(T_V - 1)d} - \epsilon\sqrt{(T_V - 1)d}$$

$$\geq d - 3\epsilon\sqrt{T_V d}$$

Thus, putting the analysis in the two cases together, as long as $T_V \geq 1$

$$\forall j \in V \qquad \left\langle \boldsymbol{\delta}_j, \sum_{i \in V} \boldsymbol{\delta}_i \right\rangle \geq d - 3\epsilon\sqrt{T_V d},$$

with probability at least $1 - 4T_V \exp\left(-c\frac{\epsilon^2}{G^4}\right)$.

**Proof of eq. (9)** Next, we apply lemma 3 again to the inner product of two vectors of i.i.d standard normal random variables:

$$\forall j \in U \qquad p\left(\left|\left\langle \boldsymbol{\delta}_j, \frac{1}{\sqrt{T_V}}\sum_{i \in V} \boldsymbol{\delta}_i \right\rangle\right| \geq \epsilon\sqrt{d}\right) < 2\exp(-c\epsilon^2/G^4).$$

By union bound over $U$

$$p\left[\forall j \in U \qquad \left(\left|\left\langle \boldsymbol{\delta}_j, \frac{1}{\sqrt{T_V}}\sum_{i \in V} \boldsymbol{\delta}_i \right\rangle\right| \geq \epsilon\sqrt{d}\right)\right] < 2T_u \exp(-c\epsilon^2/G^4).$$

Thus, with probability at least $1 - 2T_u \exp\left(-c\frac{\epsilon^2}{G^4}\right)$, the following holds, concluding the proof

$$\forall j \in U \qquad \left|\left\langle \boldsymbol{\delta}_j, \frac{1}{\sqrt{T_V}}\sum_{i \in V} \boldsymbol{\delta}_i \right\rangle\right| \leq \epsilon\sqrt{d}.$$

$\square$

**Lemma 6.** *Let $\{\mathbf{x}_i, \mathbf{y}_i\}_{i \leq n}$ be a collection of $d$ dimensional covariates $\mathbf{x}_i$ and label $\mathbf{y}_i$ sampled according to $p_\rho$ in eq. (1). The covariates $\mathbf{x}_i = [\pm B\mathbf{y}_i, \mathbf{y}_i\boldsymbol{\delta}_i]$, where $+B$ in the middle coordinate for $i \in S_{shortcut}$ and $-B$ for $i \in S_{leftover}$. The dual formulation of the following norm-minimization problem*

$$\mathbf{w}_{stable} = \arg\min_{\mathbf{w}} \quad \mathbf{w}_y^2 + \mathbf{w}_z^2 + \|\mathbf{w}_e\|^2$$

$$s.t. \; i \in S_{shortcut} \quad w_y + Bw_z + \mathbf{w}_e^\top \mathbf{y}_i\boldsymbol{\delta}_i > 1$$

$$s.t. \; i \in S_{leftover} \quad w_y - Bw_z + \mathbf{w}_e^\top \mathbf{y}_i\boldsymbol{\delta}_i > 1$$

$$\mathbf{w}_y \geq B\mathbf{w}_z$$

*is the following with $\zeta^\top = [-B, 1, \mathbf{0}^{d-2}]$,*

$$\max_{\lambda \geq 0, \nu \geq 0} -\frac{1}{4}\|\zeta\nu + X^\top\lambda\|^2 + \mathbf{1}^\top\lambda, \tag{10}$$

*where $X$ is a matrix with $\mathbf{y}_i\mathbf{x}_i$ as its rows.*

*Proof.* We use Lagrange multipliers $\lambda \in \mathbb{R}^n, \nu \in \mathbb{R}$ to absorb the constraints and then use strong duality. Letting $\zeta^\top = [-B, 1, \mathbf{0}^{d-2}]$, $X$ be a matrix where the $i$th row is $\mathbf{x}_i\mathbf{y}_i$,

$$\min_{\mathbf{w}} \quad \|\mathbf{w}\|^2 \qquad \text{s.t.} \qquad X\mathbf{w} - \mathbf{1} \geq 0 \qquad \zeta^\top\mathbf{w} \geq 0$$

has the same solution as

$$\max_{\lambda \geq 0, \nu \geq 0} \min_{\mathbf{w}} \quad \|\mathbf{w}\|^2 - (X\mathbf{w} - \mathbf{1})^\top\lambda - \nu\zeta^\top\mathbf{w} \tag{11}$$

Now, we solve the inner minimization to write the dual problem only in terms of $\lambda, \nu$. Solving the inner minimization involves solving a quadratic program, which is done by setting its gradient to zero,

$$\nabla_{\mathbf{w}}\left(\|\mathbf{w}\|^2 - (X\mathbf{w} - \mathbf{1})^\top\lambda - \nu\zeta^\top\mathbf{w}\right) = 2\mathbf{w} - X^\top\lambda - \nu\zeta = 0$$
$$\implies \mathbf{w} = \frac{1}{2}(\zeta\nu + X^\top\lambda)$$

Substituting $\mathbf{w} = \frac{1}{2}(\zeta\nu + X^\top\lambda)$ in eq. (11)

$$\|\mathbf{w}\|^2 - (X\mathbf{w} - \mathbf{1})^\top\lambda - \nu\zeta^\top\mathbf{w} =$$
$$\frac{1}{4}\|\zeta\nu + X^\top\lambda\|^2 - \left(\frac{1}{2}(X(\zeta\nu + X^\top\lambda) - \mathbf{1})^\top\lambda - \frac{1}{2}\nu\zeta^\top(\zeta\nu + X^\top\lambda)\right)$$
$$= \frac{1}{4}\|\zeta\nu + X^\top\lambda\|^2 - \left(\frac{1}{2}(X(\zeta\nu + X^\top\lambda) - \mathbf{1})^\top\lambda - \frac{1}{2}\nu^2\|\zeta\|^2 - \frac{1}{2}\nu\zeta^\top X^\top\lambda\right)$$
$$= \frac{1}{4}\|\zeta\nu + X^\top\lambda\|^2 - \frac{1}{2}(X(X^\top\lambda))^\top\lambda - \frac{1}{2}(X(\zeta\nu))^\top\lambda + \mathbf{1}^\top\lambda - \frac{1}{2}\nu^2\|\zeta\|^2 - \frac{1}{2}\nu\zeta^\top X^\top\lambda$$
$$= \frac{1}{4}\|\zeta\nu + X^\top\lambda\|^2 - \left(\frac{1}{2}(X(X^\top\lambda))^\top\lambda + \frac{1}{2}\nu^2\|\zeta\|^2 + \nu\zeta^\top X^\top\lambda\right) + \mathbf{1}^\top\lambda$$
$$= \frac{1}{4}\|\zeta\nu + X^\top\lambda\|^2 - \left(\frac{1}{2}(X^\top\lambda)^\top X^\top\lambda + \frac{1}{2}\nu^2\|\zeta\|^2 + \nu\zeta^\top X^\top\lambda\right) + \mathbf{1}^\top\lambda$$
$$= \frac{1}{4}\|\zeta\nu + X^\top\lambda\|^2 - \frac{1}{2}\left(\|X^\top\lambda\|^2 + \|\nu\zeta\|^2 + 2\nu\zeta^\top X^\top\lambda\right) + \mathbf{1}^\top\lambda$$
$$= \frac{1}{4}\|\zeta\nu + X^\top\lambda\|^2 - \frac{1}{2}\|\zeta\nu + X^\top\lambda\|^2 + \mathbf{1}^\top\lambda$$
$$= -\frac{1}{4}\|\zeta\nu + X^\top\lambda\|^2 + \mathbf{1}^\top\lambda$$

$\square$

## A.2 Shortcut learning in max-margin classification

We repeat the DGP from the linear perception task in eq. (1) here.

$$\mathbf{y} \sim \text{Rad}, \quad \mathbf{z} \sim \begin{cases} p_\rho(\mathbf{z} = y \mid \mathbf{y} = y) = \rho \\ p_\rho(\mathbf{z} = -y \mid \mathbf{y} = y) = (1 - \rho) \end{cases}, \quad \boldsymbol{\delta} \sim \mathcal{N}(0, \mathbf{I}^{d-2}), \quad \mathbf{x} = [B * \mathbf{z}, \mathbf{y}, \boldsymbol{\delta}]. \tag{12}$$

**Theorem 1.** *Let $\mathbf{w}^*$ be the max-margin predictor on $n$ training samples from eq. (12) with a leftover group of size $k$. There exist constants $C_1, C_2, N_0 > 0$ such that*

$$\forall \text{ integers } k \in \left(0, \frac{n}{10}\right) \tag{13}$$

$$\forall \; d \geq C_1 k \log(3n), \tag{14}$$

$$\forall \; B > C_2 \sqrt{d/k}, \tag{15}$$

with probability at least $1 - \frac{1}{3n}$ over draws of the training data, it holds that $B\mathbf{w}_z^* > \mathbf{w}_y^*$.

Before giving the proof of theorem 1, we first give the corollary showing overparameterization is not necessary for theorem 1 to hold.

**Corollary 1.** *For all $n > N_0$ — where the constant $N_0$ is from theorem 1 — with scalar $\tau \in (0, 1)$ such that the dimension $d = \tau n < n$, theorem 1 holds.*

$$\forall k \leq n \times \min\left\{\frac{1}{10}, \frac{\tau}{C_1 \log 3n}\right\},$$

*a linear model trained via default-ERM yields a predictor $\mathbf{w}^*$ such that $B\mathbf{w}_z^* > \mathbf{w}_y^*$.*

*Proof.* We show that for a range of $k$, for all $n \geq N_0$ theorem 1 holds for some $d < n$. Note that theorem 1 holds for $n \geq N_0, d = C_1 k \log(3n)$ and

$$\forall k < \frac{n}{10}.$$

Setting $d \leq \tau n$ for some $\tau \in (0, 1)$ such that $d < n$ means that theorem 1 holds if

$$C_1 k \log(3n) = d \leq \tau n \implies k \leq \frac{\tau n}{C_1 \log(3n)}.$$

Absorbing this new upper bound into the requirements on $k$ for theorem 1 to hold, we get that for any scalar $n > N_0, \tau \in (0, 1), d = \tau n$, theorem 1 holds for

$$\forall k < n \times \min\left\{\frac{1}{10}, \frac{\tau}{C_1 \log(3n)}\right\}.$$

In turn, even though $d < n$, a linear model trained via default-ERM converges in direction to a max-margin classifier such that $\mathbf{w}^*$ with $B\mathbf{w}_z^* > \mathbf{w}_y^*$. $\qquad\square$

*Proof.* (of theorem 1) We consider two norm-minimization problems over $\mathbf{w}$, one under constraint $\mathbf{w}_y \geq B\mathbf{w}_z$ and another under $\mathbf{w}_y < B\mathbf{w}_z$. We show that the latter achieves lower norm and therefore, max-margin will achieve solutions $\mathbf{w}_y < B\mathbf{w}_z$. The two minimization problems are as follows:

$$
\begin{aligned}
\mathbf{w}_{\text{stable}} = \arg\min_{\mathbf{w}} \quad & w_y^2 + w_z^2 + \|\mathbf{w}_e\|^2 \\
\text{s.t. } i \in S_{\text{shortcut}} \quad & w_y + Bw_z + \mathbf{w}_e^\top \mathbf{y}_i \boldsymbol{\delta}_i > 1 \\
\text{s.t. } i \in S_{\text{leftover}} \quad & w_y - Bw_z + \mathbf{w}_e^\top \mathbf{y}_i \boldsymbol{\delta}_i > 1 \\
& \mathbf{w}_y \geq B\mathbf{w}_z
\end{aligned}
\qquad
\begin{aligned}
\mathbf{w}_{\text{shortcut}} = \arg\min_{\mathbf{w}} \quad & w_y^2 + w_z^2 + \|\mathbf{w}_e\|^2 \\
\text{s.t. } i \in S_{\text{shortcut}} \quad & w_y + Bw_z + \mathbf{w}_e^\top \mathbf{y}_i \boldsymbol{\delta}_i > 1 \\
i \in S_{\text{leftover}} \quad & w_y - Bw_z + \mathbf{w}_e^\top \mathbf{y}_i \boldsymbol{\delta}_i > 1 \\
& \mathbf{w}_y < B\mathbf{w}_z
\end{aligned}
$$

$$\text{(16)} \qquad\qquad\qquad\qquad\qquad\qquad\qquad\qquad \text{(17)}$$

From eq. (16), any $\mathbf{w}$ that satisfy the constraints of the dual maximization problem will lower bound the value of the optimum of the primal, $\|\mathbf{w}_{\text{stable}}\|^2 \geq W_{\text{stable}}$. From the eq. (17), substituting a guess in $\mathbf{w}_{\text{shortcut}}$ that satisfies the constraints yields an upper bound, $\|\mathbf{w}_{\text{shortcut}}\|^2 \leq W_{\text{shortcut}}$. The actual computation of the bounds $W_{\text{shortcut}}, W_{\text{stable}}$ is in lemmas 7 and 8 which are proved in appendix A.3 and appendix A.4 respectively. We reproduce the lemmas here for convenience.

**Lemma.** (7) *Consider the following optimization problem from eq. (16) where $n$ samples of $\mathbf{x}_i, \mathbf{y}_i$ come from eq. (1) where $\mathbf{x}_i \in \mathbf{R}^d$:*

$$
\begin{aligned}
\mathbf{w}_{stable} = \arg\min_{\mathbf{w}} \quad & w_y^2 + w_z^2 + \|\mathbf{w}_e\|^2 \\
\text{s.t. } i \in S_{shortcut} \quad & w_y + Bw_z + \mathbf{w}_e^\top \mathbf{y}_i \boldsymbol{\delta}_i > 1 \\
\text{s.t. } i \in S_{leftover} \quad & w_y - Bw_z + \mathbf{w}_e^\top \mathbf{y}_i \boldsymbol{\delta}_i > 1 \\
& \mathbf{w}_y \geq B\mathbf{w}_z
\end{aligned}
\tag{18}
$$

Let $k = |S_{leftover}| > 1$. Then, for a fixed constant $c = \frac{1}{(2e)^2}$, with any scalar $\epsilon < \sqrt{d}$, with probability at least $1 - 2\exp(-c\epsilon^2/G^4)$ and $\forall$ integers $M \in \left[1, \lfloor \frac{n}{2k} \rfloor\right]$,

$$\|\mathbf{w}_{stable}\|^2 \geq W_{stable} = \frac{1}{4 + \frac{(\sqrt{d}+\epsilon)^2}{2Mk}}.$$

**Lemma.** **(8)** *Consider the following optimization problem from* eq. (16) *where $n$ samples of $\mathbf{x}_i, \mathbf{y}_i$ come from* eq. (1) *where $\mathbf{x}_i \in \mathbf{R}^d$:*

$$
\begin{aligned}
\mathbf{w}_{shortcut} = \arg\min_{\mathbf{w}} \quad & w_y^2 + w_z^2 + \|\mathbf{w}_e\|^2 \\
s.t. \ i \in S_{shortcut} \quad & w_y + Bw_z + \mathbf{w}_e^\top \mathbf{y}_i \boldsymbol{\delta}_i > 1 \\
i \in S_{leftover} \quad & w_y - Bw_z + \mathbf{w}_e^\top \mathbf{y}_i \boldsymbol{\delta}_i > 1 \\
& \mathbf{w}_y < B\mathbf{w}_z
\end{aligned}
\tag{19}
$$

Let $k = |S_{leftover}| \geq 1$. Then, for a fixed constant $c = \frac{1}{(2e)^2}$, with any scalar $\epsilon < \frac{1}{3}\sqrt{\frac{d}{k}} < \sqrt{d}$, with probability at least $1 - 2(2k + (n-k) + 1)\exp(-c\frac{\epsilon^2}{G^4})$, for $\gamma = \frac{2}{d - 4\epsilon\sqrt{kd}}$,

$$\|\mathbf{w}_{shortcut}\|^2 \leq W_{shortcut} = \gamma^2 k (\sqrt{d} + \epsilon)^2 + \frac{\left(1 + \gamma\epsilon\sqrt{dk}\right)^2}{B^2}$$

Together, the lemmas say that for any $\forall$ integers $M \in \left[1, \lfloor \frac{n}{2k} \rfloor\right]$ and $\epsilon < \frac{1}{3}\sqrt{\frac{d}{k}}$, with probability $\geq 1 - 2\exp(-c\epsilon^2/G^4)$

$$\|\mathbf{w}_{stable}\|^2 \geq W_{stable} = \frac{1}{4 + \frac{(\sqrt{d}+\epsilon)^2}{2Mk}}.$$

and with probability at least $1 - 2(2k + (n-k) + 1)\exp(-c\frac{\epsilon^2}{G^4})$, for $\gamma = \frac{2}{d - 4\epsilon\sqrt{kd}} > 0$,

$$\|\mathbf{w}_{shortcut}\|^2 \leq W_{shortcut} = \gamma^2 k (\sqrt{d} + \epsilon)^2 + \frac{\left(1 + \gamma\epsilon\sqrt{dk}\right)^2}{B^2}$$

First, we choose $\epsilon^2 = 2\frac{G^4}{c}\log(3n)$. This gives us the probability with which these bounds hold: as $k < 0.1n$ we have $k + 2 < \frac{n}{2}$ and

$$
\begin{aligned}
1 - 2(2k + (n-k) + 2)\exp(-c\frac{\epsilon^2}{G^4}) &= 1 - 2(n + k + 2)\exp(-2\log(3n)) \\
&\geq 1 - 2(\frac{3n}{2})\exp(-2\log(3n)) \\
&= 1 - \exp(-2\log(3n) + \log(3n)) \\
&= 1 - \exp(-\log(3n)) \\
&= 1 - \frac{1}{3n}.
\end{aligned}
$$

Next, we will instantiate the parameter $M$ and set the constants $C_1, C_2$ and the upper bound on $k$ in theorem 1 to guarantee the following eq. (separation inequality):

$$W_{shortcut} = \gamma^2 k (\sqrt{d} + \epsilon)^2 + \frac{\left(1 + \gamma\epsilon\sqrt{dk}\right)^2}{B^2} \quad < \quad \frac{1}{4 + \frac{(\sqrt{d}+\epsilon)^2}{2Mk}} = W_{stable},$$

(separation inequality)

which then implies that $\|\mathbf{w}_{shortcut}\|^2 < \|\mathbf{w}_{stable}\|^2$, concluding the proof.

**Invoking the conditions in theorem 1 and setting the upper bound on $k$.** We will keep the $\epsilon$ as is for simplicity of reading but invoke the inequalities satisfied by $\log(3n)$ from theorem 1:

$$\exists \text{ constant } C_1, \qquad d \geq C_1 k \log(3n).$$

Now we let $C_1 = 2\frac{G^4}{cC^2}$ for a constant $C \in \left(0, \frac{1}{3}\right)^3$, such that

$$\epsilon^2 = 2\frac{G^4}{c}\log(3n) < C^2\frac{d}{k} \implies \epsilon < C\sqrt{\frac{d}{k}} \text{ and } \epsilon\sqrt{kd} < Cd. \tag{20}$$

We next find a $C \in \left(0, \frac{1}{3}\right)$ such that eq. (separation inequality) holds with $M = 5$, which upper bounds $k$:

$$M < \frac{n}{2k} \implies \frac{k}{n} < \frac{1}{2M} = \frac{1}{10} \implies k < \frac{n}{10}.$$

**Simplifying $W_{\text{shortcut}}$ and $W_{\text{stable}}$.** To actually show $W_{\text{shortcut}} < W_{\text{stable}}$ in eq. (separation inequality), we compare a simplified strict upper bound on the LHS $W_{\text{shortcut}}$ and a simplified strict lower bound on the RHS $W_{\text{stable}}$

For the simplification of the RHS $W_{\text{stable}}$ of eq. (separation inequality), we will use the fact that $d \geq 2\frac{G^4}{cC^2}\log(3n)k$. Given the assumption $n > N_0$, choosing $N_0$ to be an integer such that $\log(3N_0) \geq \frac{40cC^2}{G^4}$ means that $\log(3n) > \frac{40cC^2}{G^4}$ and we have

$$\frac{d}{k} > 80 \implies \frac{d}{10k} > 8 \implies \frac{1}{2}\frac{d}{10k} > 4 \tag{21}$$

which gives us, for $M = 5$,

$$W_{\text{stable}} = \frac{1}{4 + \frac{(\sqrt{d}+\epsilon)^2}{2Mk}} \tag{22}$$

$$= \frac{1}{4 + \frac{(\sqrt{d}+\epsilon)^2}{10k}} \tag{23}$$

$$\geq \frac{1}{\frac{3}{2}\frac{(\sqrt{d}+\epsilon)^2}{10k}} \qquad \left\{4 < \frac{1}{2}\frac{d}{10k} < \frac{1}{2}\frac{(\sqrt{d}+\epsilon)^2}{10k} \text{ from eq. (21)}\right\} \tag{24}$$

$$= \frac{20k}{3(\sqrt{d}+\epsilon)^2} \tag{25}$$

$$\geq \frac{20k}{3(\sqrt{d}+C\frac{\sqrt{d}}{\sqrt{k}})^2} \qquad \left\{\epsilon < \frac{C\sqrt{d}}{\sqrt{k}} \text{ from eq. (20)}\right\} \tag{26}$$

$$= \frac{20k}{3(1+\frac{C}{\sqrt{k}})^2 d} \tag{27}$$

$$> \frac{20k}{3(1+C)^2 d} \qquad \{k \geq 1\} \tag{28}$$

Now, we produce a simpler upper bound on the first part of the LHS of eq. (separation inequality): recalling that $\gamma = \frac{2}{d - 4\epsilon\sqrt{kd}}$, and substituting in the upper bounds on $\epsilon$,

$$\gamma^2 k(\sqrt{d}+\epsilon)^2 = \left(\frac{2(\sqrt{d}+\epsilon)}{d - 4\epsilon\sqrt{kd}}\right)^2 k$$

$$< 4\left(\frac{(\sqrt{d}+C\sqrt{\frac{d}{k}})}{d - 4Cd}\right)^2 k \qquad \left\{\epsilon < \frac{C\sqrt{d}}{\sqrt{k}} \text{ from eq. (20)}\right\}$$

$$= \frac{4k}{d}\left(\frac{(1+\frac{C}{\sqrt{k}})}{1 - 4C}\right)^2$$

---

[3]The $\frac{1}{3}$ comes from requiring that $\epsilon < \frac{1}{3}\sqrt{\frac{d}{k}}$ from lemma 8.

$$\leq \frac{4k}{d}\left(\frac{1+C}{1-4C}\right)^2, \qquad \{k \geq 1\} \tag{29}$$

Next is a simpler upper bound on the second part of the LHS of eq. (separation inequality). Again with $\gamma = \frac{2}{d-4\epsilon\sqrt{kd}}$,

$$\frac{\left(1+\gamma\epsilon\sqrt{dk}\right)^2}{B^2} = \frac{\left(1+\frac{2\epsilon\sqrt{dk}}{d-4\epsilon\sqrt{kd}}\right)^2}{B^2}$$

$$\leq \frac{\left(1+\frac{2Cd}{d-4Cd}\right)^2}{B^2}$$

$$= \frac{\left(1+\frac{2C}{1-4C}\right)^2}{B^2}$$

Now setting

$$B > \sqrt{2}\frac{\left(1+\frac{2C}{1-4C}\right)}{\sqrt{\frac{4k}{d}\left(\frac{1+C}{1-4C}\right)}}$$

gives the lower bound on $B$ from theorem 1:

$$B > C_2\sqrt{\frac{d}{k}}, \qquad \text{where} \quad C_2 = \frac{\left(1+\frac{2C}{1-4C}\right)}{\sqrt{2}\left(\frac{1+C}{1-4C}\right)} = \frac{(1-2C)}{\sqrt{2}(1+C)}.$$

Formally,

$$B > C_2\sqrt{\frac{d}{k}} \implies \frac{\left(1+\frac{2C}{1-4C}\right)^2}{B^2} < \frac{1}{2}\left(\sqrt{\frac{4k}{d}\left(\frac{1+C}{1-4C}\right)}\right)^2 = \frac{1}{2}\frac{4k}{d}\left(\frac{1+C}{1-4C}\right)^2. \tag{30}$$

By combining the upper bound from eq. (30) and the upper bound from eq. (29), we get an upper bound on the whole of the LHS of eq. (separation inequality), which in turn provides an upper bound on $W_{\text{shortcut}}$:

$$W_{\text{shortcut}} = \gamma^2 k(\sqrt{d}+\epsilon)^2 + \frac{\left(1+\gamma\epsilon\sqrt{dk}\right)^2}{B^2} < \frac{3}{2}\frac{4k}{d}\left(\frac{(1+C)}{1-4C}\right)^2 \leq \frac{3}{2}\frac{4k}{d}\left(\frac{(1+C)}{1-4C}\right)^2,$$

because $k \geq 1$. Note the upper bound is strict.

**Concluding the proof.** Now, we show that a $C$ exists such that the following holds, which implies $W_{\text{shortcut}} < W_{\text{stable}}$, which in turn implies eq. (separation inequality) and the proof concludes:

$$W_{\text{shortcut}} < \frac{3}{2}\frac{4k}{d}\left(\frac{(1+C)}{1-4C}\right)^2 \leq \frac{20k}{3(1+C)^2 d} < W_{\text{stable}}.$$

The above inequality holds when

$$6\left(\frac{(1+C)}{1-4C}\right)^2 \leq \frac{20}{3(1+C)^2} \iff (1+C)^2 - \sqrt{\frac{10}{9}}(1-4C) \leq 0.$$

The right hand side holds when the quadratic equation $(1+C)^2 - \sqrt{\frac{10}{9}}(1-4C)$ is non-positive, which holds between the roots of the equation. The equation's positive solution is

$$C = \frac{-3+\sqrt{10}}{3+2\sqrt{10}+\sqrt{5(8+3\sqrt{10})}} \approx 0.008.$$

Setting $C$ to this quantity satisfies the requirement that $C \in (0, \frac{1}{3})$.

Thus, a $C$ exists such that eq. (separation inequality) holds which concludes the proof of theorem 1 for the following constants and constraints implied by $C$ and $M = 5$:

$$C_2 = \frac{(1 - 2C)}{\sqrt{2}(1 + C)} \qquad C_1 = 2\frac{G^4}{cC^2} \qquad k < \frac{n}{10},$$

where $G$ is the $\psi_2$-norm of a standard normal r.v. and $c$ is the absolute constant from the Bernstein inequality in lemma 2. $\qquad\square$

### A.3 Lower bounding the norm of solutions that rely more on the stable feature

**Lemma 7.** *Consider the following optimization problem from eq. (16) where $n$ samples of $\mathbf{x}_i, \mathbf{y}_i$ come from eq. (1) where $\mathbf{x}_i \in \mathbf{R}^d$:*

$$
\begin{aligned}
\mathbf{w}_{stable} = \arg\min_{\mathbf{w}} \quad & w_y^2 + w_z^2 + \|\mathbf{w}_e\|^2 \\
s.t. \ i \in S_{shortcut} \quad & w_y + Bw_z + \mathbf{w}_e^\top \mathbf{y}_i \boldsymbol{\delta}_i > 1 \\
s.t. \ i \in S_{leftover} \quad & w_y - Bw_z + \mathbf{w}_e^\top \mathbf{y}_i \boldsymbol{\delta}_i > 1 \\
& \mathbf{w}_y \geq B\mathbf{w}_z
\end{aligned}
\tag{31}
$$

*Let $k = |S_{leftover}| > 1$. Then, for a fixed constant $c = \frac{1}{(2e)^2}$, with any scalar $\epsilon < \sqrt{d}$, with probability at least $1 - 2\exp(-c\epsilon^2/G^4)$ and $\forall$ integers $M \in \left[1, \lfloor \frac{n}{2k} \rfloor\right]$,*

$$\|\mathbf{w}_{stable}\|^2 \geq W_{stable} = \frac{1}{4 + \frac{(\sqrt{d}+\epsilon)^2}{2Mk}}.$$

*Proof.* By lemma 6, the dual of eq. (16) is the following for $\zeta = [-B, 1, \mathbf{0}^{d-2}]$ and $X$ is an $n \times d$ matrix with rows $\mathbf{y}_i \mathbf{x}_i$:

$$\max_{\lambda \geq 0, \nu \geq 0} -\frac{1}{4}\|\zeta\nu + X^\top \lambda\|^2 + \mathbf{1}^\top \lambda \tag{32}$$

Now by duality

$$\|\mathbf{w}_{stable}\|^2 \geq \max_{\lambda \geq 0, \nu \geq 0} -\frac{1}{4}\|\zeta\nu + X^\top \lambda\|^2 + \mathbf{1}^\top \lambda,$$

which means any feasible candidate to eq. (32) gives a lower bound on $\|\mathbf{w}_{stable}\|^2$.

**Feasible Candidates for $\lambda, \nu$.** We now define a set $U \subset [n]$, and let $\lambda_i = \frac{\alpha}{|U|} > 0$ for $i \in U$ and $0$ otherwise. For $M \in (1, \lfloor \frac{n}{2k} \rfloor]$, we take $2Mk$ samples from the training data to be included in $U$. Formally,

$$U = S_{leftover} \cup (2M - 1)k \text{ a random samples from } S_{shortcut},$$

which gives the size $|U| = 2Mk$. Then, we let $\nu = \alpha\frac{2(M-1)}{2M} > 0$.

Note that for the above choice of $\lambda$, $X^\top \lambda$ is a sum of the rows from $U$ scaled by $\frac{\alpha}{|U|}$. Adding up $k$ rows from $S_{leftover}$ and $k$ rows from $S_{shortcut}$ cancels out the $B$s and, so in the $B$ is accumulated $|U| - 2k = 2(M - 1)k$ times, and so

$$X^\top \lambda = \left[\alpha * \frac{|U| - 2k}{|U|}B, \alpha, \frac{\alpha}{|U|}\sum_{i \in U} \boldsymbol{\delta}_i\right] = \left[\alpha B\frac{2(M-1)}{2M}, \alpha, \frac{\alpha}{|U|}\sum_{i \in U} \boldsymbol{\delta}_i\right].$$

As $\lambda$ has $\frac{\alpha}{|U|}$ on $|U|$ elements and $0$ otherwise, $\lambda^\top \mathbf{1} = \alpha$

As we set $\nu = \alpha\frac{2(M-1)}{2M}$,

$$\nu\zeta + X^\top \lambda = \left[-\alpha B\frac{2(M-1)}{2M} + \alpha\frac{2(M-1)}{2M}B, \alpha\frac{2(M-1)}{2M} + \alpha, 0 + \frac{\alpha}{|U|}\sum_{i} \boldsymbol{\delta}_i\right]$$

(33)

$$= \left[ 0 \quad , \alpha \left( 1 + \frac{2(M-1)}{2M} \right), \quad \frac{\alpha}{|U|} \sum_i \boldsymbol{\delta}_i \right] \tag{34}$$

$$\implies \|\zeta\nu + X^\top \lambda\|^2 = \left\| \left[ 0, \alpha \left( 1 + \frac{2(M-1)}{2M} \right), \frac{\alpha}{|U|} \sum_{i \in U} \boldsymbol{\delta}_i \right] \right\|^2 \tag{35}$$

$$= \alpha^2 \left\| \left[ 0, \left( 1 + \frac{2(M-1)}{2M} \right), \frac{1}{|U|} \sum_{i \in U} \boldsymbol{\delta}_i \right] \right\| \tag{36}$$

For the chosen values of $\nu, \lambda$ the value of the objective in eq. (32) is

$$\frac{-\alpha^2}{4} \left\| \left[ 0, \left( 1 + \frac{2(M-1)}{2M} \right), \frac{1}{|U|} \sum_{i \in U} \boldsymbol{\delta}_i \right] \right\|^2 + \alpha \tag{37}$$

Letting

$$\Gamma = \left\| \left[ 0, \left( 1 + \frac{2(M-1)}{2M} \right), \frac{1}{|U|} \sum_{i \in U} \boldsymbol{\delta}_i \right] \right\|^2,$$

the objective is of the form $\alpha - \frac{\alpha^2 \Gamma}{4}$. To maximize with respect to $\alpha$, setting the derivative of the objective w.r.t $\alpha$ to 0 gives:

$$1 - \frac{2\alpha\Gamma}{4} = 0 \implies \alpha = \frac{2}{\Gamma} \implies \alpha - \frac{\alpha^2 \Gamma}{4} = \frac{2}{\Gamma} - \frac{4}{\Gamma^2}\frac{\Gamma}{4} = \frac{1}{\Gamma}.$$

This immediately gives us

$$\|\mathbf{w}_{\text{stable}}\|^2 \geq \frac{1}{\Gamma},$$

and we lower bound this quantity by upper bounding $\Gamma$.

By concentration of gaussian norm as in lemma 4, with probability at least $1 - 2\exp(-c\frac{\epsilon^2}{G^4})$

$$\left\| \frac{1}{|U|} \sum_{i \in U} \boldsymbol{\delta}_i \right\| = \frac{1}{\sqrt{|U|}} \left\| \frac{1}{\sqrt{|U|}} \sum_{i \in U} \boldsymbol{\delta}_i \right\| \leq \frac{1}{\sqrt{|U|}} (\sqrt{d} + \epsilon).$$

In turn, recalling that $|U| = 2Mk$

$$\Gamma \leq \left( \frac{(2(M-1) + 2M)}{2M} \right)^2 + \left( \frac{\sqrt{d} + \epsilon}{\sqrt{|U|}} \right)^2 < 4 + \left( \frac{\sqrt{d} + \epsilon}{\sqrt{|U|}} \right)^2 \leq 4 + \frac{\left( \sqrt{d} + \epsilon \right)^2}{2Mk}$$

The upper bound on $\Gamma$ gives the following lower bound on $\|\mathbf{w}_{\text{stable}}\|^2$:

$$\|\mathbf{w}_{\text{stable}}\|^2 \geq \frac{1}{\Gamma} \geq \frac{1}{4 + \frac{(\sqrt{d}+\epsilon)^2}{2Mk}}$$

$\square$

### A.4 Upper bounding the norm of solutions that rely more on the shortcut.

**Lemma 8.** *Consider the following optimization problem from eq. (16) where $n$ samples of $\mathbf{x}_i, \mathbf{y}_i$ come from eq. (1) where $\mathbf{x}_i \in \mathbf{R}^d$:*

$$
\begin{aligned}
\mathbf{w}_{shortcut} = \arg\min_{\mathbf{w}} \quad & w_y^2 + w_z^2 + \|\mathbf{w}_e\|^2 \\
s.t. \ i \in S_{shortcut} \quad & w_y + Bw_z + \mathbf{w}_e^\top \mathbf{y}_i \boldsymbol{\delta}_i > 1 \\
i \in S_{leftover} \quad & w_y - Bw_z + \mathbf{w}_e^\top \mathbf{y}_i \boldsymbol{\delta}_i > 1 \\
& \mathbf{w}_y < B\mathbf{w}_z
\end{aligned}
\tag{38}
$$

*Let $k = |S_{leftover}| \geq 1$. Then, for a fixed constant $c = \frac{1}{(2e)^2}$, with any scalar $\epsilon < \frac{1}{3}\sqrt{\frac{d}{k}} < \sqrt{d}$, with probability at least $1 - 2(2k + (n-k) + 1)\exp(-c\frac{\epsilon^2}{G^4})$, for $\gamma = \frac{2}{d - 4\epsilon\sqrt{kd}}$,*

$$\|\mathbf{w}_{shortcut}\|^2 \leq W_{shortcut} = \gamma^2 k(\sqrt{d} + \epsilon)^2 + \frac{\left(1 + \gamma\epsilon\sqrt{dk}\right)^2}{B^2}$$

*Proof.* Let $k = |S_{\text{leftover}}|$. The candidate we will evaluate the objective for is

$$\mathbf{w} = \left[\frac{\beta}{B}, 0, \gamma \sum_{j \in S_{\text{leftover}}} \mathbf{y}_j \boldsymbol{\delta}_j\right]. \tag{39}$$

**High-probability bounds on the margin achieved by the candidate and norm of w**  The margins on the shortcut group and the leftover group along with the constraints are as follows:

$$\forall j \in S_{\text{shortcut}} \quad m_j = 0 + B * \frac{\beta}{B} + \left\langle \mathbf{y}_j \boldsymbol{\delta}_j, \gamma \sum_{i \in S_{\text{leftover}}} \mathbf{y}_i \boldsymbol{\delta}_i \right\rangle \geq 1$$

$$\forall j \in S_{\text{leftover}} \quad m_j = 0 - B * \frac{\beta}{B} + \left\langle \mathbf{y}_j \boldsymbol{\delta}_j, \gamma \sum_{i \in S_{\text{leftover}}} \mathbf{y}_i \boldsymbol{\delta}_i \right\rangle \geq 1. \tag{40}$$

Due to the standard normal distribution being isotropic, and $\mathbf{y}_j \in \{-1, 1\}$, $\mathbf{y}_j \boldsymbol{\delta}_j$ has the same distribution as $\boldsymbol{\delta}_j$. Then, we apply lemma 5 with $V = S_{\text{leftover}}, U = S_{\text{shortcut}}$ — which means $T_v = k$ and $T_u = (n - k)$ — to bound the margin terms in eq. (40) and $\|\mathbf{w}\|^2$ with probability at least

$$1 - 2(2k + (n - k) + 2)\exp(-c\frac{\epsilon^2}{G^4}).$$

Applying the bound in eq. (9) in lemma 5 between a sum of vectors and a different i.i.d vector,

$$\forall j \in S_{\text{shortcut}} \quad \left|\left\langle \mathbf{y}_j \boldsymbol{\delta}_j, \gamma \sum_{i \in S_{\text{leftover}}} \mathbf{y}_i \boldsymbol{\delta}_i \right\rangle\right| \leq \gamma\epsilon\sqrt{kd} \tag{41}$$

Applying the bound in eq. (8) from lemma 5

$$\forall j \in S_{\text{leftover}} \quad \left\langle \mathbf{y}_j \boldsymbol{\delta}_j, \gamma \sum_{i \in S_{\text{leftover}}} \mathbf{y}_i \boldsymbol{\delta}_i \right\rangle \geq \gamma\left(d - 3\epsilon\sqrt{kd}\right) \tag{42}$$

The margin constraints on the shortcut and leftover from eq. (40) respectively imply

$$\beta - \gamma\epsilon\sqrt{dk} \geq 1 \qquad -\beta + \gamma\left(d - 3\epsilon\sqrt{kd}\right) \geq 1$$

We choose $\beta = 1 + \gamma\epsilon\sqrt{dk}$, which implies an inequality that $\gamma$ has to satisfy the following, which is due to $d - 3\epsilon\sqrt{kd} > 0$,

$$-(1 + \gamma\epsilon\sqrt{dk}) + \gamma\left(d - 3\epsilon\sqrt{kd}\right) \geq 1 \implies \gamma \geq \frac{2}{d - 4\epsilon\sqrt{kd}}$$

Now, we choose

$$\gamma = \frac{2}{d - 4\epsilon\sqrt{kd}}.$$

**Computing the upper bound on the value of the objective in the primal problem in eq. (17)**  The feasible candidate's norm $\|\mathbf{w}\|^2$ is an upper bound on the solution's norm $\|\mathbf{w}_{\text{shortcut}}\|^2$ and so

$$\|\mathbf{w}_{\text{shortcut}}\|^2 \leq \|\mathbf{w}\|^2 = \frac{1}{B^2}\beta^2 + \left\|\gamma \sum_{j \in S_{\text{leftover}}} \mathbf{y}_j \boldsymbol{\delta}_j\right\|^2 = \gamma^2 k \left\|\frac{1}{\sqrt{k}} \sum_{j \in S_{\text{leftover}}} \boldsymbol{\delta}_j\right\|^2 + \frac{\beta^2}{B^2}$$

By lemma 5 which we invoked,

$$\left\| \frac{1}{\sqrt{k}} \sum_{j \in S_{\text{leftover}}} \boldsymbol{\delta}_j \right\|^2 \leq (\sqrt{d} + \epsilon)^2.$$

To conclude the proof, substitute $\beta = 1 + \gamma \epsilon \sqrt{dk}$ and get the following upper bound with $\gamma = \frac{2}{d - 3\epsilon\sqrt{kd}}$:

$$\|\mathbf{w}_{\text{shortcut}}\|^2 \leq \gamma^2 k (\sqrt{d} + \epsilon)^2 + \frac{\beta^2}{B^2} = \gamma^2 k (\sqrt{d} + \epsilon)^2 + \frac{\left(1 + \gamma \epsilon \sqrt{dk}\right)^2}{B^2}.$$

$\square$

## A.5  Concentration of $k$ and intuition behind theorem 1

**Concentration of $k$ around $(1 - \rho)n$.**  Denote the event that the $i$th sample lies in the leftover group as $I_i$: then $E[I_i] = 1 - \rho$ and the leftover group size is $k = \sum_i I_i$. Hoeffding's inequality (Theorem 2.2.6 in [Vershynin, 2018]) shows that for any $t > 0$, $k$ is at most $(1 - \rho)n + t\sqrt{n}$ with probability at least $1 - \exp(-2t^2)$:

$$p\left(k - (1 - \rho)n > t\sqrt{n}\right) = p\left(\sum_i (I_i - (1 - \rho)) > t\sqrt{n}\right) = p\left(\sum_i (I_i - E[I_i]) > t\sqrt{n}\right) \leq \exp(-2t^2).$$

Letting $\rho = 0.9 + \sqrt{\frac{\log 3n}{n}}$ and $t = \sqrt{\log 3n}$, gives us

$$\begin{aligned}
p\left(k - (1 - \rho)n > t\sqrt{n}\right) &= p\left(k - 0.1n + \sqrt{n \log 3n} > \sqrt{\log 3n}\sqrt{n}\right) \\
&= p\left(k - 0.1n > 0\right) \\
&\leq \exp(-2t^2) \\
&= \exp(-2 \log 3n). \\
&= \left(\frac{1}{3n}\right)^2 \\
&< \frac{1}{3n}
\end{aligned}$$

To connect $\rho$ to shortcut learning due to max-margin classification, we take a union bound of the event that $k < 0.1n$, which occurs with probability at least $1 - \frac{1}{3n}$ and theorem 1 which occurs with probability at least $1 - \frac{1}{3n}$. This union bound guarantees that with probability at least $1 - \frac{2}{3n}$ over sampling the training data, max-margin classification on $n$ training samples from eq. (1) relies more on the shortcut feature if $\rho$ is above a threshold; and this threshold converges to 0.9 at the rate of $\sqrt{\log 3n/n}$.

## A.6  Bumpy losses improve ERM in the under-parameterized setting

**Theorem 2.** *Consider $n$ samples of training data from* DGP *in eq. (1) with $d < n$. Consider a linear classifier $f_\theta(\mathbf{x}) = \mathbf{w}^\top \mathbf{x}$ such that for all samples in the training data $\mathbf{y}_i \mathbf{w}^\top \mathbf{x}_i = b$ for any $b \in (0, \infty)$. With probability 1 over draws of samples, $\mathbf{w} = [0, b, 0^{d-2}]$.*

*Proof.* Letting $X$ be the matrix where each row is $\mathbf{y}_i \mathbf{x}_i$, the theorem statement says the solution $\mathbf{w}^*$

$$X\mathbf{w}^* = b\mathbf{1} \tag{43}$$

First, split $\mathbf{w}^* = [w_z^*, w_y^*, \mathbf{w}_{-y}^*]$. Equation (43) says that the margin of the model on any sample satisfies

$$\mathbf{y}(\mathbf{w}^*)^\top \mathbf{x} = w_y^* \mathbf{y}^2 + w_z^* \mathbf{yz} + \mathbf{y}(\mathbf{w}_{-y}^*)^\top \boldsymbol{\delta} = b \qquad \Longrightarrow \qquad \mathbf{y}(\mathbf{w}_{-y}^*)^\top \boldsymbol{\delta} = b - w_y^* \mathbf{y}^2 - w_z^* \mathbf{yz}$$

We collect these equations for the whole training data by splitting $X$ into columns: denoting $Y, Z$ as vectors of $\mathbf{y}_i$ and $\mathbf{z}_i$ and using $\cdot$ to denote element wise operation, split $X$ into columns that correspond to $\mathbf{y}, \mathbf{z}$ and $\boldsymbol{\delta}$ respectively as $X = [Y \cdot Y \mid Y \cdot Z \mid X_\delta]$. Rearranging terms gives us

$$w_z^* Y \cdot Z + w_y^* \mathbf{1} + X_\delta \mathbf{w}_\delta^* = b\mathbf{1} \qquad \Longrightarrow \qquad X_\delta \mathbf{w}_\delta^* = (b - w_y^*)\mathbf{1} - w_z^* Y \cdot Z.$$

The elements of $Y \cdot Z$ lie in $\{-1, 1\}$ and, as the shortcut feature does not always equal the label, the elements of $Y \cdot Z$ are not all the same sign.

**Solutions do not exist when one non-zero element exists in** $(b - w_y^*)\mathbf{1} - w_z^* Y \cdot Z$  By definition of $\mathbf{w}^*$

$$X_\delta \mathbf{w}_\delta^* = (b - w_y^*)\mathbf{1} - w_z^* Y \cdot Z.$$

Denote $r = (b - w_y^*)\mathbf{1} - w_z^* Y \cdot Z$. and $A = X_\delta$. Now we show that w.p. 1 solutions do not exist for the following system of linear equations:

$$Aw = r.$$

First, note that $A = X_\delta$ has $\mathbf{y}_i \boldsymbol{\delta}_i$ for rows and as $\mathbf{y}_i \perp\!\!\!\perp \boldsymbol{\delta}_i$ and $\mathbf{y}_i \in \{-1, 1\}$, each vector $\mathbf{y}_i \boldsymbol{\delta}_i$ is distributed identically to a vector of independent standard Gaussian random variables. Thus, $A$ is a matrix of IID standard Gaussian random variables.

Let $U$ denote $D - 2$ indices such that the corresponding rows of $A$ form a matrix $D - 2 \times D - 2$ matrix and $r_U$ has at least one non-zero element; let $A_U$ denote the resulting matrix. Now $A_U$ is a $D - 2 \times D - 2$ sized matrix where each element is a standard Gaussian random variable. Such matrices have rank $D - 2$ with probability 1 because square singular matrices form a measure zero set under the Lebesgue measure over $\mathbf{R}^{D-2 \times D-2}$[Feng and Zhang, 2007].

We use subscript $\cdot_{-U}$ to denote all but the indices in $U$. The equation $Aw = r$ implies the following two equations:

$$A_U w = r_U \qquad\qquad A_{-U} w = r_{-U}.$$

As $A_U$ is has full rank $(D - 2)$, $A_U w = r_U$ admits a unique solution $\mathbf{w}_U^* \neq 0$ — because $r_U$ has at least one non-zero element by construction. Then, it must hold that

$$A_{-U} \mathbf{w}_U^* = r_{-U}. \tag{44}$$

For any row $v^\top \in A_{-U}$, eq. (44) implies that $v^\top \mathbf{w}^*$ equals a fixed constant. As $v$ is a vector of i.i.d standard normal random variables, $v^\top \mathbf{w}^*$ is a gaussian random variable with mean $\sum(\mathbf{w}_i^*)$ and variance $\|\mathbf{w}^*\|^2$. Then with probability 1, $v^\top \mathbf{w}^*$ will not equal a constant. Thus, w.p.1 $A_{-U} \mathbf{w}_U^* = r_{-U}$ is not satisfied, which means w.p.1 there are no solutions to $A\mathbf{w} = r$.

**Case where** $(b - w_y^*)\mathbf{1} - w_z^* Y \cdot Z$ **is zero element-wise**  As $X$ has rank $D - 2$, $X_\delta \mathbf{w}_\delta^* = 0$ only when $\mathbf{w}_\delta^* = 0$.

Each element in $(b - w_y^*)\mathbf{1} - w_z^* Y \cdot Z$ is either $b - w_y^* + w_z^*$ or $b - w_y^* - w_z^*$. Thus,

$$(b - w_y^*)\mathbf{1} - w_z^* Y \cdot Z = 0 \quad \Longrightarrow \quad \left\{ \begin{array}{l} b - w_y^* + w_z^* = 0, \\ b - w_y^* - w_z^* = 0 \end{array} \right. \tag{45}$$

Adding and subtracting the two equations on the right gives

$$2(b - w_y^*) = 0 \qquad \text{and} \qquad 2w_z^* = 0.$$

Thus, $\mathbf{w}_\delta^* = 0, w_z^* = 0, b = w_y^*$.  $\square$

# B  Appendix: further experimental details and results

## B.1  Default-ERM with $\ell_2$-regularization.

In section 3, we show default-ERM achieves zero training loss by using the shortcut to classify the shortcut group and noise to classify the leftover group, meaning the leftover group is overfit. The usual way to mitigate overfitting is via $\ell_2$-regularization, which, one can posit, may encourage models to rely on the perfect stable feature instead of the imperfect shortcut and noise.

We train the linear model from section 3 with default-ERM and $\ell_2$-regularization — implemented as weight decay in the AdamW optimizer [Loshchilov and Hutter, 2019] — on data from eq. (1) with

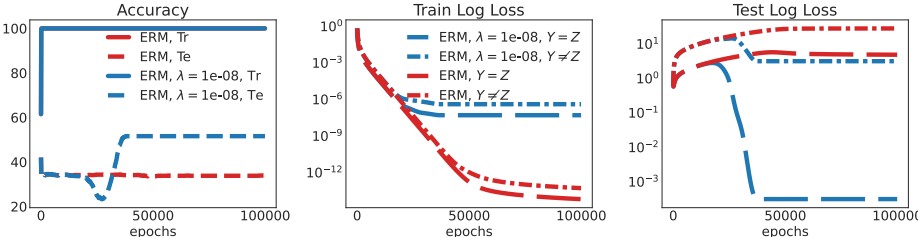

**Figure 4:** Default-ERM with $\ell_2$-regularization with a penalty coefficient of $\lambda = 10^{-8}$ achieves a test accuracy of $\approx 50\%$, outperforming default-ERM. The right panel shows that $\ell_2$-regularization leads to lower test loss on the minority group, meaning that the regularization does mitigate some overfitting. However, the difference between the shortcut and leftover test losses shows that the model still relies on the shortcut.

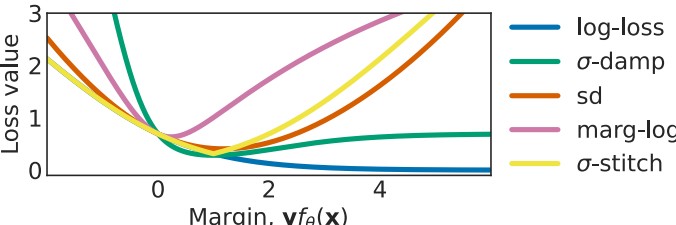

**Figure 5:** Comparing log-loss with MARG-CTRL as functions of the margin. Each MARG-CTRL loss has a "bump" which characterizes the loss function's transition from a decreasing function of the margin to an increasing one. These bumps push models to have uniform margins because the loss function's derivative after the bump is negative which discourages large margins. The hyperparameters (temperature in $\sigma$-damp or function output target in MARG-LOG.) affect the location of the bump and the slopes of the function on either side of the bump.

$d = 800, B = 10, n = 1000$. Figure 4 plots accuracy and losses for the $\ell_2$-regularized default-ERM with the penalty coefficient set to $10^{-8}$; it shows that $\ell_2$-regularization leads default-ERM to build models that only achieve $\approx 50\%$ test accuracy.

For smaller penalty coefficients, default-ERM performs similar to how it does without regularization, and for larger ones, the test accuracy gets worse than default-ERM without regularization. We give an intuitive reason for why larger $\ell_2$ penalties may lead to larger reliance on the shortcut feature. Due to the scaling factor $B = 10$ in the synthetic experiment, for a fixed norm budget, the model achieves lower loss when using the shortcut and noise compared to using the stable feature. In turn, heavy $\ell_2$-regularization forces the model to rely more on the shortcut to avoid the cost of larger weight needed by the model to rely on the stable feature and the noise.

### B.2  Margin control (MARG-CTRL)

In fig. 5, we plot the different MARG-CTRL losses along with log-loss. Each MARG-CTRL loss has a "bump" which characterizes the loss function's transition from a decreasing function of the margin to an increasing one. These bumps push models to have uniform margins because the loss function's derivative after the bump is negative which discourages large margins. The hyperparameters — like temperature in $\sigma$-damp or function output target in MARG-LOG — affect the location of the bump and the slopes of the function on either side of the bump.

### B.3  MARG-CTRL on a linear model

In fig. 6, we compare default-ERM to $\sigma$-stitch. In fig. 7 and fig. 8, compare SD and MARG-LOG respectively to default-ERM. The left panel of all figures shows that MARG-CTRL achieves better test accuracy than default-ERM, while the right most panel shows that the test loss is better on the leftover group using MARG-CTRL. Finally, the middle panel shows the effect of controlling margins

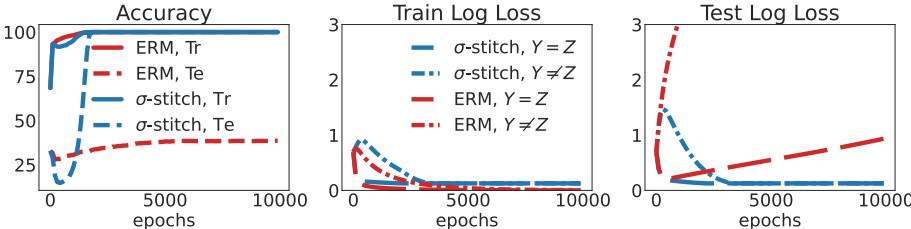

**Figure 6:** A linear trained with $\sigma$-stitch depend on the perfect stable feature to achieve perfect test accuracy, unlike default-ERM. The middle panel shows that $\sigma$-stitch does not let the loss on the training shortcut group to go to zero, unlike default-ERM, and the right panel shows the test leftover group loss is better.

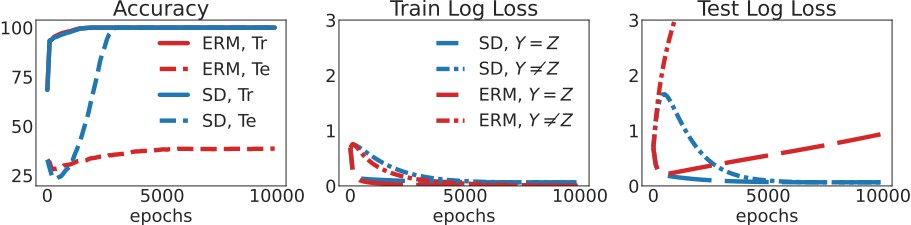

**Figure 7:** A linear model trained with SD depend on the perfect stable feature to achieve perfect test accuracy whereas default-ERM performs worse than random chance. The middle panel shows that SD does not let the loss on the training shortcut group to go to zero, unlike vanilla default-ERM, and the right panel shows the test-loss is better for the leftover group.

in training; namely, the margins on the training data do not go to $\infty$, evidenced by the training loss being bounded away from $0$. Depending on the shortcut feature leads to different margins and therefore test losses between the shortcut and leftover groups; the right panel in each plot shows that the the test losses on both groups reach similar values, meaning MARG-CTRL mitigates dependence on the shortcut. While default-ERM fails to perform better than chance ($50\%$) even after $100,000$ epochs (see fig. 1), MARG-CTRL mitigates shortcut learning within $5000$ epochs and achieves $100\%$ test accuracy.

### B.4  MARG-CTRL vs. default-ERM with a neural network

With $d = 100$ and $B = 10$ in eq. (1), we train a two layer neural network on $3000$ samples from the training distribution. The two layer neural network has a $200$ unit hidden layer that outputs a scalar. Figure 9 shows that a neural network trained via default-ERM fails to cross $50\%$ test accuracy even after $40,000$ epochs, while achieving less than $10^{-10}$ in training loss.

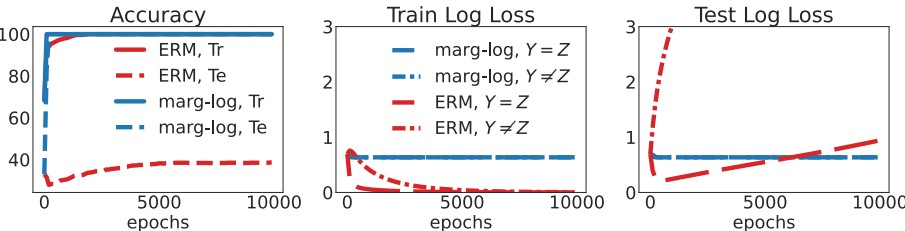

**Figure 8:** A linear model trained with MARG-LOG depend on the perfect stable feature to achieve perfect test accuracy whereas default-ERM performs worse than random chance. The middle panel shows that MARG-LOG does not let the loss on the training shortcut group to go to zero, unlike default-ERM, and the right panel shows the test-loss is better for the leftover group.

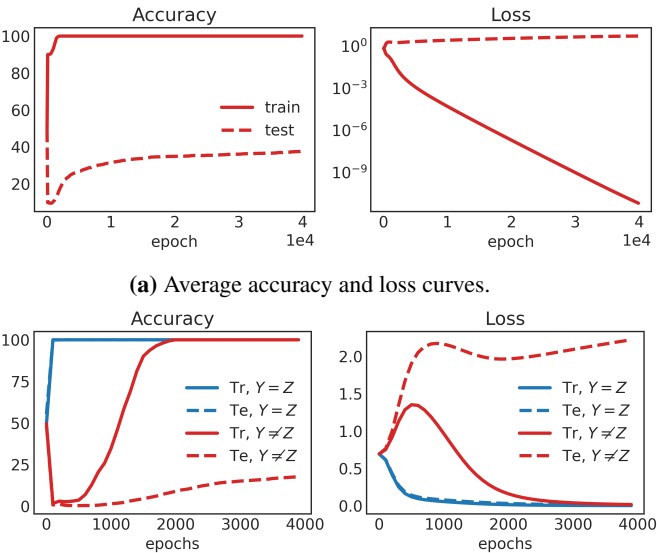

**(a)** Average accuracy and loss curves.

**(b)** Accuracy and loss on shortcut and leftover groups.

**Figure 9:** Training a two-layer neural network with default-ERM on data from eq. (1). The model achieves $100\%$ train accuracy but $< 40\%$ test accuracy even after $40,000$ epochs. The plot below zooms in on the first $4000$ epochs and shows that the model drives down loss on the test shortcut groups but not on the test leftover group. This shows that the model uses the shortcut to classify the shortcut group and noise for the leftover.

In fig. 10, we compare default-ERM to $\sigma$-stitch. In fig. 12 and fig. 13, compare SD and MARG-LOG respectively to default-ERM. The left panel of all figures shows that MARG-CTRL achieves better test accuracy than default-ERM, while the right most panel shows that the test loss is better on the leftover group using MARG-CTRL. Finally, the middle panel shows the effect of controlling margins in training; namely, the margins on the training data do not go to $\infty$, evidenced by the training loss being bounded away from $0$.

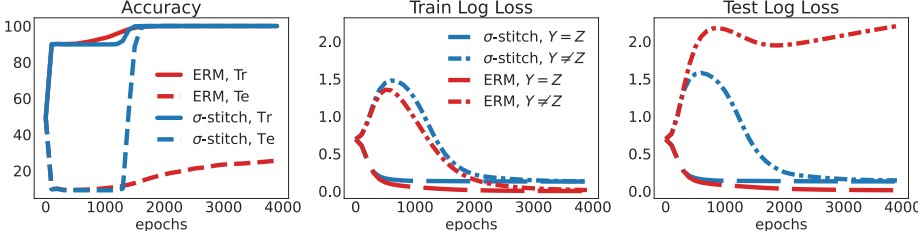

**Figure 10:** A neural network trained with $\sigma$-stitch depend on the perfect stable feature to achieve perfect test accuracy, unlike default-ERM. The middle panel shows that $\sigma$-stitch does not let the loss on the training shortcut group to go to zero, unlike default-ERM, and the right panel shows the test leftover group loss is better.

### B.5 Spectral decoupling for a linear model on the linear DGP in eq. (1).

We first show that a linear classifier trained with SD achieves $100\%$ test accuracy while default-ERM performs worse than chance on the test data; so, SD builds models with more dependence on the stable perfect feature, compared to Empirical Risk minimization (ERM). Next, we outline the assumptions for the gradient starvation (GS) regime from Pezeshki et al. [2021] and then instantiate it for a linear model under the data generating process in eq. (1), showing that the assumptions for the GS-regime are violated.

Figure 7 shows the results of training a linear model with SD on training data of size 1000 sampled as per eq. (1) from $p_{\rho=0.9}$ with $d = 300$; the test data also has a 1000 samples but comes from $p_{\rho=0.1}$.

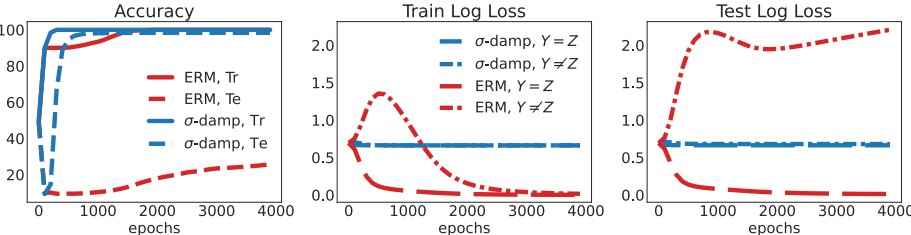

**Figure 11:** A neural network trained with $\sigma$-damp depend on the perfect stable feature to achieve perfect test accuracy whereas default-ERM performs worse than random chance. The middle panel shows that $\sigma$-damp does not let the loss on the training shortcut group to go to zero, unlike vanilla default-ERM, and the right panel shows the test-loss is better for the leftover group.

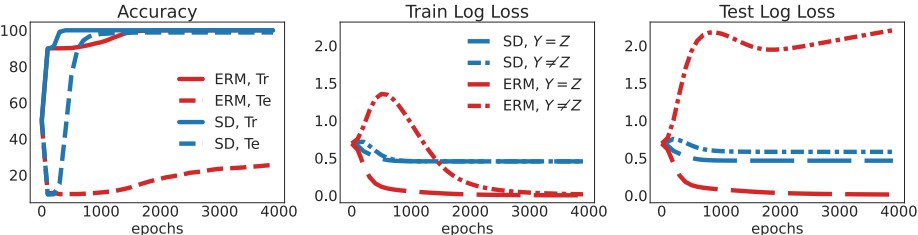

**Figure 12:** A neural network trained with SD depend on the perfect stable feature to achieve perfect test accuracy whereas default-ERM performs worse than random chance. The middle panel shows that SD does not let the loss on the training shortcut group to go to zero, unlike vanilla default-ERM, and the right panel shows the test-loss is better for the leftover group.

Figure 7 shows that SD builds models with improved dependence on the perfect stable feature, as compared to ERM, to achieve $100\%$ test accuracy.

### B.5.1 The linear example in Equation (1) violates the gradient starvation regime.

**Background on Pezeshki et al. [2021].** With the aim of explaining why ERM-trained neural networks depend more on one feature over a more informative one, Pezeshki et al. [2021] derive solutions to $\ell_2$-regularized logistic regression in the NTK; they let the regularization coefficient be small enough for the regularized solution to be similar in direction to the unregularized solution. Given $n$ samples $\mathbf{y}^i, \mathbf{x}^i$, let $\mathbf{Y}$ be a diagonal matrix with the labels on its diagonal, $\mathbf{X}$ be a matrix with $\mathbf{x}^i$ as its rows, and $\hat{\mathbf{y}}(\mathbf{X}, \theta) = f_\theta(\mathbf{X})$ be the $n$-dimensional vector of function outputs where each element is $\hat{\mathbf{y}}^i = f_\theta(\mathbf{x}^i)$. In gradient-based training in the NTK regime, the vector of function outputs of the network with parameters $\theta$ can be approximated as $\hat{\mathbf{y}} = \Phi_0 \theta$, where $\Phi_0$ is the neural-tangent-random-feature (NTRF) matrix at initialization:

$$\Phi_0 = \frac{\partial \hat{\mathbf{y}}(\mathbf{X}, \theta_0)}{\partial \theta_0}$$

To define the features, the strength (margin) of each feature, and how features appear in each sample, Pezeshki et al. [2021] compute the singular value decomposition (SVD) of the NTRF $\Phi_0$ multiplied by the diagonal-label matrix $\mathbf{Y}$:

$$\mathbf{Y}\Phi_0 = \mathbf{U}\mathbf{S}\mathbf{V}^\top. \tag{46}$$

The rows of $\mathbf{V}$ are features, the diagonal elements of $\mathbf{S}$ are the strengths of each feature and the $i$th row of $\mathbf{U}$ denotes how each feature appears in the NTRF representation of the $i$th sample.

To study issues with the solution to $\ell_2$-regularized logistic regression, Pezeshki et al. [2021] define the gradient starvation (GS) regime. Under the GS regime, they assume $\mathbf{U}$ is a perturbed identity matrix that is also unitary: for a small constant $\delta << 1$, such a matrix has all diagonal elements $\sqrt{1 - \delta^2}$ and the rest of the elements are of the order $\delta$ such that the rows have unit $\ell_2$-norm.

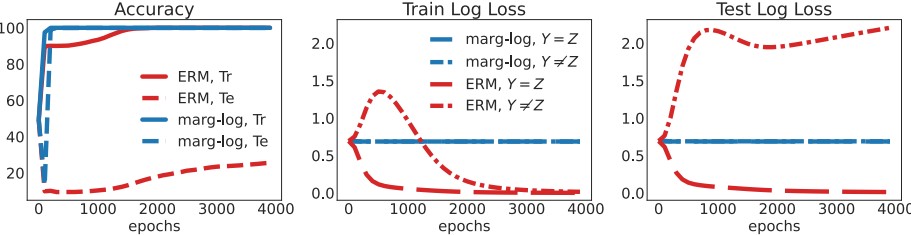

**Figure 13:** A neural network trained with MARG-LOG depend on the perfect stable feature to achieve perfect test accuracy whereas default-ERM performs worse than random chance. The middle panel shows that MARG-LOG does not let the loss on the training shortcut group to go to zero, unlike default-ERM, and the right panel shows the test-loss is better for the leftover group.

**The GS regime is violated in eq. (1).** When $f_\theta$ is linear, $f_\theta(\mathbf{x}) = \theta^\top \mathbf{x}$, the NTRF matrix is

$$\frac{\partial \hat{\mathbf{y}}(\mathbf{X}, \theta_0)}{\partial \theta_0} = \frac{\partial \mathbf{X}\theta_0}{\partial \theta_0} = \mathbf{X}.$$

In this case, let us look at an implication of $\mathbf{U}$ being a perturbed identity matrix that is also unitary, as Pezeshki et al. [2021] assume. With $(\mathbf{u}^i)^\top$ as the $i$th row of $\mathbf{U}$, the transpose of $i$th sample can be written as $(\mathbf{x}^i)^\top = (\mathbf{u}^i)^\top \mathbf{SV}$. Pezeshki et al. [2021] assume that $\delta << 1$ in that the off-diagonal terms of $\mathbf{U}$ are small perturbations such that off-diagonal terms of $\mathbf{U}(\mathbf{S}^2 + \lambda \mathbf{I})\mathbf{U}^\top$ have magnitude much smaller than 1, meaning that the terms $|(\mathbf{u}^i)^\top \mathbf{S}^2 (\mathbf{u}^j) + \lambda| << 1$ for $i \neq j$ and positive and small $\lambda << 1$.

Then,

$$|\mathbf{y}^i \mathbf{y}^j (\mathbf{x}^i)^\top \mathbf{x}^j| = |(\mathbf{x}^i)^\top \mathbf{x}^j| \tag{47}$$

$$= |(\mathbf{u}^i)^\top \mathbf{SV}^\top \mathbf{VSu}^j| \tag{48}$$

$$= |(\mathbf{u}^i)^\top \mathbf{S}^2 \mathbf{u}^j| \tag{49}$$

$$<< 1 \tag{50}$$

In words, this means that any two samples $\mathbf{x}^i, \mathbf{x}^j$ are nearly orthogonal. Now, for samples from eq. (1), for any $i, j$ such that $\mathbf{z}^j = \mathbf{z}^i$ and $\mathbf{y}^i = \mathbf{y}^j$,

$$\left|(\mathbf{x}^i)^\top \mathbf{x}^j\right| = \left|B^2 \mathbf{z}^i \mathbf{z}^j + \mathbf{y}^i \mathbf{y}^j + (\delta^i)^\top \delta^j\right| \geq |100 + 1 + (\delta^i)^\top \delta^j| \tag{51}$$

As $\delta$ are isotropic Gaussian vectors, around half the pairs $i, j$ will have $(\delta^i)^\top \delta^j > 0$ meaning $\left|(\mathbf{x}^i)^\top \mathbf{x}^j\right| > 101$. This lower bound implies that $\mathbf{U}$ is not a perturbed identity matrix for samples from eq. (1). This violates the setup of the gradient starvation regime from [Pezeshki et al., 2021].

Thus, the linear DGP in eq. (1) does not satisfy the conditions for the GS regime that is proposed in [Pezeshki et al., 2021]. The GS regime blames the coupled learning dynamics for the different features as the cause for default-ERM-trained models depending more on the less informative feature. Pezeshki et al. [2021] derive spectral decoupling (SD) to avoid coupling the training dynamics, which in turn can improve a model's dependence on the perfect feature. SD adds a penalty to the function outputs which Pezeshki et al. [2021] show decouples training dynamics for the different features as defined by the NTRF matrix:

$$\ell_{\text{SD}}(\mathbf{y}, f_\theta(\mathbf{x})) = \log(1 + \exp(\mathbf{y}f_\theta)) + \lambda |f_\theta(\mathbf{x})|^2$$

As eq. (1) lies outside the GS regime, the success of SD on data from eq. (1) cannot be explained as a consequence of avoiding the coupled training dynamics in the GS regime Pezeshki et al. [2021]. However, looking at SD as MARG-CTRL, the success of SD, as in fig. 7, is explained as a consequence encouraging uniform margins.

**An example of perfect test accuracy even with dependence on the shortcut.** In fig. 14, we train a linear model with default-ERM on data from eq. (1), showing that even when shortcut dependence

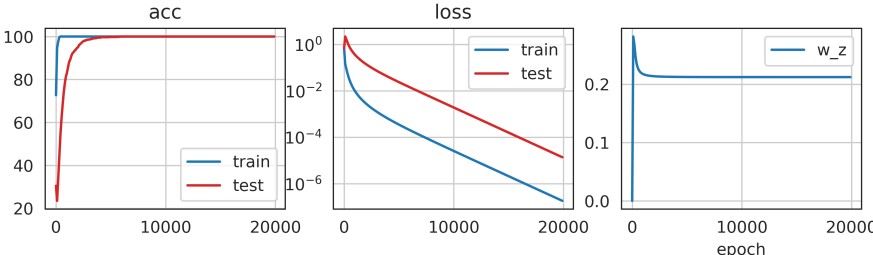

**Figure 14:** With $d = 200$ and $n = 1000$, a linear classifier can still depend on the shortcut feature and achieve $100\%$ test accuracy. Nagarajan et al. [2021] consider linearly separable data and formalize geometric properties of the data that make max-margin classifiers give non-zero weight to the shortcut feature ($\mathbf{w}_z > 0$). In their example, it is unclear when $\mathbf{w}_z > 0$ leads to poor accuracy in the leftover group because Nagarajan et al. [2021] do not separate the model's dependence on the stable feature from the dependence on noise. The example here gives an example where $\mathbf{w}_z > 0$ but test accuracy is $100\%$. demonstrating that guarantees on test leftover group error require comparing $\mathbf{w}_y$ and $\mathbf{w}_z$; the condition $\mathbf{w}_z > 0$ alone is insufficient.

is non-zero, test leftover group accuracy can be $100\%$. Nagarajan et al. [2021] consider linearly separable data and formalize geometric properties of the data that make max-margin classifiers give non-zero weight to the shortcut feature ($\mathbf{w}_z > 0$). In their example, it is unclear when $\mathbf{w}_z > 0$ leads to poor accuracy in the leftover group because Nagarajan et al. [2021] do not separate the model's dependence on the stable feature from the dependence on noise. The example in fig. 14 gives an example where $\mathbf{w}_z > 0$ but test accuracy is $100\%$, demonstrating that guarantees on test leftover group error require comparing $\mathbf{w}_y$ and $\mathbf{w}_z$; the condition $\mathbf{w}_z > 0$ alone is insufficient. In contrast, theorem 1 characterizes cases where leftover group accuracy is worse than random even without overparameterization.

## B.6 Experimental details

### B.6.1 Background on Just Train Twice (JTT) and Correct-n-Contrast (CNC)

**JTT** Liu et al. [2021] develop JTT with the aim of building models robust to subgroup shift, where the mass of disjoint subgroups of the data changes between training and test times. To work without training group annotations, JTT assumes ERM builds models with high worst-group error. With this assumption, JTT first builds an "identification" model via ERM to pick out samples that are misclassified due to model's dependence on the shortcut. Then, JTT trains a second model again via ERM on the same training data with the loss for the misclassified samples upweighted (by constant $\lambda$). As Liu et al. [2021] point out, the number of epochs to train the identification model and the upweighting constant are hyperparameters that require tuning using group annotations. As Liu et al. [2021], Zhang et al. [2022] show that JTT and CNC outperforms LFF and other two-stage shortcut-mitigating methods ([Zhang et al., 2022]), so we do not compare against them.

**Correct-n-Contrast (CNC)** In a fashion similar to JTT, the first stage of CNC is to train a model with regularized ERM to predict based on spurious attributes, i.e. shortcut features. Zhang et al. [2022] develop a contrastive loss to force the model to have similar representations across samples that share a label but come from different groups (approximately inferred by the first-stage ERM model). Formally, the first-stage model is used to approximate the spurious attributes in one of two ways: 1) predict the label with the model, 2) cluster the representations into as many clusters as there are classes, and then use the cluster identity. The latter technique was first proposed in [Sohoni et al., 2020]. For an anchor sample $(\mathbf{y}^i, \mathbf{x}^i)$ of label $\mathbf{y} = y$, positive samples $P_i$ are those than have the same label but have the predicted spurious attribute is a different value: $\hat{z} \neq y$. Negatives $N_i$ are those that have a different label but the spurious attribution is the same: $\hat{z} = y$. For a temperature parameter $\tau$ and representation function $r_\theta$, the per-sample contrastive loss for CNC is:

$$\ell_{cont}(r_\theta, i) = \mathbb{E}_{\mathbf{x}^p \sim P_i} \left[ -\log \frac{\exp\left(r_\theta(\mathbf{x}^i)^\top r_\theta(\mathbf{x}^p)/\tau\right)}{\sum_{n \in N_i} \exp\left(r_\theta(\mathbf{x}^i)^\top r_\theta(\mathbf{x}^n)/\tau\right) + \sum_{p \in P_i} \exp\left(r_\theta(\mathbf{x}^i)^\top r_\theta(\mathbf{x}^p)/\tau\right)} \right].$$

The samples $i$ are called *anchors*. For a scalar $\lambda$ to trade off between contrastive and predictive loss, the overall per-sample loss in the second-stage in CNC is

$$\lambda \ell_{cont}(r_\theta, i) + (1 - \lambda)\ell_{log-loss}(\mathbf{y}^i w^\top r_\theta(\mathbf{x}^i)).$$

**CNC uses hyperparameters informed by dataset-specific empirical results from prior work.** The original implementation of CNC from Zhang et al. [2022] uses specific values of first-stage hyperparameters like weight decay and early stopping epoch for each dataset by using empirical results from prior work [Sagawa et al., 2020a, Liu et al., 2021]. The prior work finds weight-decay and early stopping epoch which lead default-ERM models to achieve low test worst-group accuracy, implying that the model depends on the spurious attribute. This means the first-stage models built in CNC are pre-selected to pay attention to the spurious attributes. For example, [Zhang et al., 2022] point out that the first-stage model they use for Waterbirds predicts the spurious feature with an accuracy of $94.7\%$.

Without using dataset-specific empirical results from prior work, choosing LR and WD requires validating through the whole CNC procedure. We let CNC use the same LR and WD for both stages and then validate the choice using validation performance of the second-stage model. This choice of hyperparameter validation leads to a similar number of validation queries for all methods that mitigate shortcuts.

### B.6.2 Training details

**Variants of MARG-CTRL to handle label imbalance.** The three datasets that we use in our experiments — Waterbirds, CelebA, and Civilcomments — all have an imbalanced (non-uniform) marginal distribution over the label; for each dataset,

$$\max_{\text{class} \in \{-1, 1\}} p(\mathbf{y} = \text{class}) > 0.75.$$

When there is sufficiently large imbalance, restricting the margins on all samples could bias the training to reduce loss on samples in the most-frequent class first and overfit on the rest of the samples. This could force a model to predict the most frequent class for all samples, resulting in high worst-group error.

To prevent such a failure mode, we follow [Pezeshki et al., 2021] and define variants of $\sigma$-damp, MARG-LOG, and $\sigma$-stitch that have either 1) different maximum margins for different classes or 2) different per-class loss values for the same margin value. Mechanically, these variants encourage uniform margins within each class, thus encouraging the model to rely less on the shortcut feature. We give the variants here for labels taking values in $\{-1, 1\}$:

1. With per-class temperatures $T_{-1}, T_1 > 0$ the variant of $\sigma$-damp is

    with $f_\theta = w_f^\top r_\theta(\mathbf{x})$,
    $$\ell_{\sigma\text{-damp}}(\mathbf{y}, f_\theta) = \ell_{log}\left[T_{\mathbf{y}} * 1.278\mathbf{y}f_\theta\left(1 - \sigma\left(1.278 * \mathbf{y}f_\theta\right)\right)\right]$$

    The 1.278 comes in to make sure the maximum input to log-loss occurs at $f_\theta = 1$. However, due to the different temperatures $T_1 \neq T_{-1}$, achieving the same margin on all samples produces lower loss on the class with the larger temperature.

2. With per-class temperatures $T_{-1}, T_1 > 0$ the variant of $\sigma$-stitch is

    with $f_\theta = w_f^\top r_\theta(\mathbf{x})$,
    $$\ell_{\sigma\text{-stitch}}(\mathbf{y}f_\theta) = \ell_{log}\left(T_{\mathbf{y}}\left[\quad \mathbf{1}[\mathbf{y}f_\theta(\mathbf{x}) < 1] \times \mathbf{y}f_\theta(\mathbf{x}) + \mathbf{1}[\mathbf{y}f_\theta(\mathbf{x}) > 1] \times (2 - \mathbf{y}f_\theta(\mathbf{x}))\;\right]\right)$$

3. With per-class function output targets $\gamma_{-1}, \gamma_1 > 0$ the variant of MARG-LOG is

    with $f_\theta = w_f^\top r_\theta(\mathbf{x})$,
    $$\ell_{\text{MARG-LOG}}(\mathbf{y}f_\theta) = \ell_{log}(\mathbf{y}f_\theta) + \lambda \log(1 + |f_\theta - \gamma_{\mathbf{y}}|^2).$$

These per-class variants are only for training; at test time, the predicted label is $\texttt{sign}(f_\theta)$.

**Details of the vision and language experiments.** We use the same datasets from Liu et al. [2021], downloaded via the scripts in the code from [Idrissi et al., 2022]; see [Idrissi et al., 2022] for sample sizes and the group proportions. For the vision datasets, we finetune a resnet50 from Imagenet-pretrained weights and for Civilcomments, we finetune a BERT model.

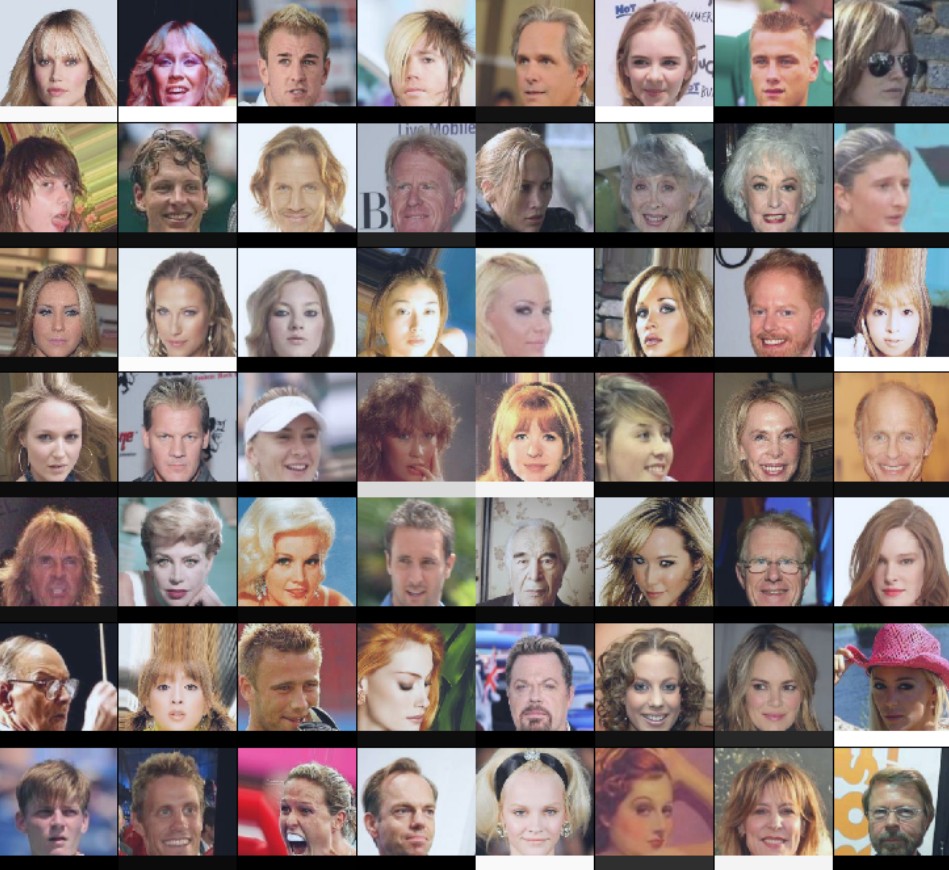

**Figure 15:** Images mis-classified by a model trained on CelebA data with equal group sizes, i.e. without a shortcut. Samples with blonde as the true label have a white strip at the bottom while samples with non-blonde as the true label have a black strip at the bottom. The figure demonstrates that many images with blonde people in the image have the non-blonde label, thus demonstrating label noise. For example, see a blonde man in the first row that is labelled non-blonde and a non-blonde lady in the third row that is lablled blonde. Yet, MARG-CTRL improves over ERM for many LR and WD combinations; see fig. 16.

**Optimization details.** For all methods and datasets, we tune over the following weight decay (WD) parameters: $10^{-1}, 10^{-2}, 10^{-3}, 10^{-4}$ For the vision datasets, we tune learning rate (LR) over $10^{-4}, 10^{-5}$ and for CivilComments, we tune over $10^{-5}, 10^{-6}$. For CivilComments, we use the AdamW optimizer while for the vision datasets, we use the Adam optimizer; these are the standard optimizers for their respective tasks [Puli et al., 2022, Gulrajani and Lopez-Paz, 2021]. We use a batch size of 128 for both CelebA and Waterbirds, and train for 20 and 100 epochs respectively. For CivilComments we train for 10 epochs with a batch size of 16.

**Per-method Hyperparameters.** Like in [Pezeshki et al., 2021], the per-class temperatures $T_{-1}, T_1$ for $\sigma$-damp and $\sigma$-stitch, and the function output targets $\gamma_{-1}, \gamma_1$ for MARG-LOG are hyperparameters that we tune using the worst-group accuracy or label-balanced average accuracy computed on the validation dataset, averaged over 2 seeds.

1. For $\sigma$-stitch, we select from $T_{-1} \in \{1, 2\}$ and $T_1 \in \{2, 4, 8, 12\}$ such that $T_1 > T_{-1}$.

2. For $\sigma$-damp, we search over $T_{-1} \in \{1, 2\}$ and $T_1 \in \{2, 4\}$ such that $T_1 > T_{-1}$.

3. For SD and MARG-LOG, we search over $\gamma_{-1} \in \{-1, 0, 1\}$ and $\gamma_1 \in \{1, 2, 2.5, 3\}$ for the image datasets and $\gamma_1 \in \{1, 2\}$ for the text dataset, and the penalty coefficient is set to be $\lambda = 0.1$

4. For JTT, we search over the following parameters: the number of epochs $T \in \{1, 2\}$ for CelebA and Civilcomments and $T \in \{10, 20, 30\}$ for Waterbirds, and the upweighting constant $\lambda \in \{20, 50, 100\}$ for the vision datasets and $\lambda \in \{4, 5, 6\}$ for Civilcomments.

5. For CNC, we search over the same hyperparameter as [Zhang et al., 2022] : the temperature in $\tau \in \{0.05, 0.1\}$, the contrastive weight $\lambda \in \{0.5, 0.75\}$, and the gradient accumulation steps $s \in \{32, 64\}$. For the language task in Civilcomments, we also try one additional $s = 128$.

## B.7 MARG-CTRL improves over default-ERM on CelebA even without the stable feature being perfect.

CelebA is a perception task in that the stable feature is the color of the hair in the image. But unlike the synthetic experiments, MARG-CTRL does not achieve a $100\%$ test accuracy on CelebA. We investigated this and found that CelebA in fact has some label noise.

We trained a model via the MARG-CTRL method $\sigma$-damp on CelebA data with no shortcut; this data is constructed by subsampling the groups to all equal size, (5000 samples). This achieves a test worst-group accuracy of $89\%$. We visualized the images that were misclassified by this model and found that many images with blond-haired people were classified as having non-blonde hair. Figure 15 shows 56 misclassified images where samples with blonde as the true label have a white strip at the bottom while samples with non-blonde as the true label have a black strip at the bottom. The figure shows that images with blonde people can have the non-blonde label, thus demonstrating label noise. Thus, MARG-CTRL improves over ERM even on datasets like CelebA where the stable features do not determine the label.

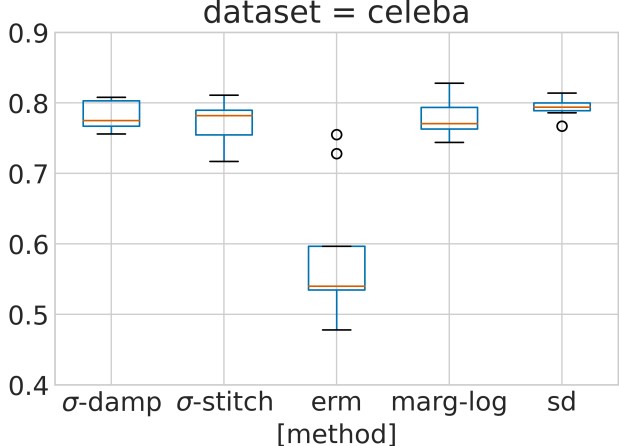

**Figure 16:** Test worst-group accuracy on CelebA of default-ERM and MARG-CTRL for different values of LR and WD. Default-ERM's performance changes more with LR and WD than MARG-CTRL, which shows that default-ERM is more sensitive than MARG-CTRL. Only 2 combinations of LR and WD improve ERM beyond a test worst-group accuracy of $60\%$, while every MARG-CTRL method achieves more than $70\%$ test worst-group accuracy for every combination of LR and WD.

## B.8 Sensitivity of ERM and MARG-CTRL to varying LR and WD

In fig. 16, we compare the test worst-group accuracy of default-ERM and MARG-CTRL on CelebA, for different values of LR and WD. There are 8 combinations of LR and WD for which ERM is run. For each combination of LR and WD, the hyperparameters of the MARG-CTRL method (values of $\lambda, T, v$) are tuned using validation group annotations, and the test worst-group accuracy corresponds to the best method hyperparameters. Default-ERM's performance changes more with LR and WD than MARG-CTRL, which shows that default-ERM is more sensitive than MARG-CTRL. Only 2 combinations of LR and WD improve ERM beyond a test worst-group accuracy of $60\%$, while every MARG-CTRL method achieves more than $70\%$ test worst-group accuracy for every combination of LR and WD.

