# OpenReview forum: "Don’t blame Dataset Shift! Shortcut Learning due to Gradients and Cross Entropy"
_NeurIPS.cc/2023/Conference — NeurIPS 2023 poster_

### Official Review · Reviewer_9QKU · 2023-07-06

**Soundness:** 3 good
**Presentation:** 3 good
**Contribution:** 3 good
**Rating:** 6
**Confidence:** 3

**Summary:**

This paper theoretecally and empirically showed that the inductive bias of default- ERM maximizing the margin causes shortcut learning in  a linear perception task.
It proposed uniform margins that leads to models that depend more on the stable than the shortcut feature and suggested loss functions encourage uniform-margin solutions.

**Strengths:**

This paper analyzes shortcut learning theoretically in terms of margin maximization.
I have not seen an analysis from this perspective before.
It also proposes the concept of uniform margin from theory and suggests a method to prevent shortcut learning.

**Weaknesses:**

The theory itself has limited applicability due to the linear model, and we do not know how well it can actually be explained in general terms in actual deep learning models.

**Questions:**

It seems that this paper forcus on a theoretical analysis in a situation like background shortcuts, where shortcut features make up a large portion of the data as shown in term B.
Learning  features with a large B preferentially is intuitively reasonable in such a simple situation; however, reading the proof in the Appendix, it seems to require a very tough proof to show that.
I would like to know where the difficulty of the proof lies, what you devised in the Appendix proof ,and where you derived the key inequality.

Shortcut features do not always occupy as much space as stable features as background features. For example, in the following paper, the percentage of partial input is much smaller.
>Overinterpretation reveals image classification model pathologies Brandon Carter, Siddhartha Jain, Jonas Mueller, David Gifford
Isn't Bz>y important for the theoretical analysis?
If Bz<y, what part of the proof is more difficult? Or what other assumptions are necessary?





**Limitations:**

The theory itself has limited applicability due to the linear model, and we do not know how well it can actually be explained in general terms in actual deep learning models.

---

> ### Author Rebuttal · Authors · 2023-08-10
>
> Thank you for your thoughtful feedback. We addressed your questions and concerns below. If any residual concerns remain, we would be glad to discuss further. If no concerns remain, we would appreciate it if you could raise your score.
>
> **[The theory itself has limited applicability due to the linear model, and we do not know how well it can actually be explained in general terms in actual deep learning models.]**
>
> The experiments in the paper empirically confirm that our theoretical insights do in fact extend to real deep neural networks: the solutions motivated by the theory — margin-control techniques — mitigate shortcuts in actual deep learning models on real-world vision and language datasets.
>
> Many works in the literature study linear models to develop insights into why deep models fail to be robust [1,2,3]. In a similar vein, we show that shortcut learning occurs in linear models despite the presence of a perfect stable feature because of max-margin classification. As default-ERM also maximizes margins in neural networks [4], the insights from our theory should extend beyond linear models which we confirm empirically.
>
> 1. Sagawa, Shiori, et al. "An investigation of why overparameterization exacerbates spurious correlations." International Conference on Machine Learning. PMLR, 2020.
> 2. Nagarajan, Vaishnavh, Anders Andreassen, and Behnam Neyshabur. "Understanding the failure modes of out-of-distribution generalization." International Conference on Learning Representations. 2020.
> 3. Wald, Yoav et al. “Malign Overfitting: Interpolation and Invariance are Fundamentally at Odds.” International Conference on Learning Representations (2023).
> 4. Lyu, Kaifeng, and Jian Li. "Gradient Descent Maximizes the Margin of Homogeneous Neural Networks." International Conference on Learning Representations. 2019.
>
> **[It seems that this paper focuses on a theoretical analysis in a situation like background shortcuts, where shortcut features make up a large portion of the data as shown in term B. Shortcut features do not always occupy as much space as stable features as background features.]**
>
> This is a great question!
>
> We would like to clarify that in the linear perception task, the shortcut is a single scalar coordinate of the covariates. With large B, models that predict with the shortcut can achieve margins greater than 1 while having a small norm: to achieve a margin > 1 on the shortcut group a linear predictor only needs 1/B weight on the shortcut feature. To achieve the same margin with the stable feature, the linear predictor needs a weight of 1.
>
> A large $B$ is analogous to shortcut pixels in images that vary dramatically (+1 to -1 for example in normalized pixel-space) when the label changes. For example, in figure 1 in the paper the reviewer links (https://arxiv.org/abs/2003.08907), classifying the image of the bird mostly requires just the blue pixels, which are part of the background sky. These few background pixels vary a lot between different background types from sky to land.
>
> In effect, what matters in shortcut learning is pixel values that vary a lot across examples. If large portions of pixels vary together from large to small, models could learn to depend on them faster but it is not necessary for all shortcuts to be large portions of the input. This intuition should help see that the results do not require the backgrounds to make up a large portion of the data.
>
> We have added this discussion to the paper.
>
> **[Learning features with a large B preferentially is intuitively reasonable in such a simple situation; however, reading the proof in the Appendix, it seems to require a very tough proof to show that. I would like to know where the difficulty of the proof lies, what you devised in the Appendix proof ,and where you derived the key inequality.]**
>
> The reason the proof is complicated is that there is no closed-form solution for the max-margin classifier, hence we need to derive a bound for the reliance on the shortcut. The key inequality is named the *separation inequality* (on line 709 in the appendix)*,* which shows that max-margin solutions that rely more on the stable feature require higher norm than max-margin solutions that rely on the shortcut feature.
>
> **[Isn't Bz>y important for theoretical analysis? If Bz<y, what part of the proof is more difficult? Or what other assumptions are necessary?]**
>
> We interpret this as the reviewer asking about the role of the magnitude of $B$, in that what default-ERM would learn if $| B z | < | y |$. It is true that $B$ needs to be a positive number greater than 1; theorem 1 shows a lower bound on $B$ that along with large enough $d, \rho$ yields shortcut learning in default-ERM.
>
> If $B<1$, default-ERM leads to models that depend more on the stable feature than the shortcut, which allows them to achieve 100% test accuracy in the linear perception task. Intuitively, this is because when $B < 1$, the model can achieve more margin on every sample by relying on the stable feature instead of the shortcut feature, for the same cost in $\ell_2$-norm.

---

> > ### Comment · Reviewer_9QKU · 2023-08-18
> > **After rebuttle**
> >
> > Thanks for replysing my questions,.
> > My quettions about the theory are totally answered and clear.
> > However, it still seems that the scope of application is limited compared to the complexity of the theory.
> > If you demonstrate the importance of the theory in the current problem setting, it would be better to explain specific examples of the problem setting used in the theory analysis, as well as confirm the findings from the theory for deep neural networks.
> > For example, after a linear approximation like NTK, I would like to know if your setting for shortcut learning could occur in real world image data.
> > Therefore, we would like to keep the score as it is now.

---

> > > ### Author Response · Authors · 2023-08-20
> > >
> > > Thank you for your response and feedback!
> > >
> > > The reviewer asks an important question: is there empirical confirmation or application of the insights from the theory beyond linear models?
> > >
> > > Extending the theory to deep neural network is an open question and an interesting one. However, the theoretical insights from the linear case lead to methods that show empirical value in the nonlinear case. The experiments in the original draft of the paper already do confirm that our insights extend to deep models and real-world data:
> > >
> > > 1. Figures 1a and 1b show shortcut learning when training a two-layer neural network on the linear perception tasks data.
> > > 2. With the insights from the theory, we develop MARG-CTRL which does mitigate shortcuts in deep networks on real-world tasks, thus validating the effectiveness of our theoretical insights in deep networks on real data
> > >
> > > Second, the reviewer asks whether our setting for shortcut learning occurs in real world image data.
> > >
> > > Our setting captures real perception tasks where the stable feature is much more informative than the shortcut: for example, CelebA has a strong stable feature that, by itself, achieves an average accuracy of 93%, while the shortcut feature alone at best achieves 84% in average accuracy. Our experiments show that MARG-CTRL mitigates shortcuts on this task.
> > >
> > > If the reviewer meant something else by "I would like to know if your setting for shortcut learning could occur in real world image data" we would be happy to discuss further.

---

### Official Review · Reviewer_PgqC · 2023-07-06

**Soundness:** 4 excellent
**Presentation:** 4 excellent
**Contribution:** 4 excellent
**Rating:** 5
**Confidence:** 3

**Summary:**

This paper provides an in-depth analysis of the phenomenon of "shortened learning" in machine learning models, especially in the context of perceptual tasks. The authors confirm that basic empirical risk minimization (ERM) methods tend to prefer models that depend on shortcut features, even when models can achieve zero loss using only stable features. They attribute this to ERM's inductive bias to maximize margins across all samples. To address this, the authors propose an alternative loss function biased inductively with a uniform margin called MARG-CTRL. The paper demonstrates that MARG-CTRL mitigates shortcut learning on multiple vision and language tasks without the use of annotations of the shortcut feature in training or even validation. They also show that MARG-CTRL performs on par or better than the more complex and costly two-step shortcut mitigation method.

**Strengths:**

-This paper presents a thorough analysis of the shortcut learning problem.
-The authors propose a novel solution, MARG-CTRL, which has been shown to effectively mitigate shortcut learning in several tasks.
-The paper is well-written and easy to understand.

**Weaknesses:**

-Although MARG-CTRL is shown to perform well across different tasks, it is not clear how MARG-CTRL perform in scenarios where the shortcut features and the stable features are highly correlated.
-The paper compares MARG-CTRL with two-stage shortcut-mitigating methods like JTT and CNC, it would be helpful to understand the specific scenarios where MARG-CTRL outperforms these methods and where it does not.

**Questions:**

-How does MARG-CTRL perform in scenarios where the shortcut features and the stable features are highly correlated? Does this affect the effectiveness of MARG-CTRL?
-What are the trade-offs involved in using MARG-CTRL versus two-stage shortcut-mitigating methods??
-Are there specific scenarios or types of tasks for which MARG-CTRL may not be as effective?

**Limitations:**

Please see the weaknesses part.

---

> ### Author Rebuttal · Authors · 2023-08-10
>
> Thank you for your thoughtful feedback. We addressed your questions and concerns below. If any residual concerns remain, we would be glad to discuss further. If no concerns remain, we would appreciate it if you could raise your score.
>
> **[How does MARG-CTRL perform in scenarios where the shortcut features and the stable features are highly correlated? Does this affect the effectiveness of MARG-CTRL?]**
>
> We interpret this as the reviewer asking about the situation where conditioning on the shortcut features leaves little randomness in the stable feature. This is already the case in the synthetic experiment; for example, conditioning on the value of the shortcut feature $X_1 = B$, the stable feature $X_2 = 1$ in 90% of the samples. As demonstrated in the paper in figures 2 and 7-10, MARG-CTRL learns the stable feature and achieves 100% test accuracy on this.
>
> In the case of perfect correlation, the case where $Z=Y$ in the data, even knowing Z is insufficient to build models that depend on the stable feature. To see this, consider the following two families of data-generating processes $p_1$ and $p_2$.
>
> Let $ p_1(Y, Z, X; \rho)$ be sampled as
>
> $$ \begin{align}
> Y ~ U({-1, 1}) , \quad
> Z = & \quad Y \text{ with probability } \rho
> \qquad \qquad
>   X = \left[Y ,Z\right].
> \\\\
> = & - Y \text{ with probability } 1-\rho
> \end{align}$$
>
> Let $ p_2(Y, Z, X; \rho)$ the same as $p_1$ except with the feature order in $X$ swapped ($X = [ Z, Y ] $ ).
>
> When Z and Y are perfectly correlated, that is $\rho=1$, $p_1 ( Y, Z, X; \rho=1) = p_2 ( Y, Z, X; \rho=1)$ meaning the observed data is the same and any model training procedure would produce the same results in distribution. However, the stable predictor for $p_1$ should place weight on the first feature of $X$, which is different from the stable predictor for $p_2$ that should place weight on the second feature of $X$.
>
> We have added this discussion to the paper.
>
> **[What are the trade-offs involved in using MARG-CTRL versus two-stage shortcut-mitigating methods?? -Are there specific scenarios or types of tasks for which MARG-CTRL may not be as effective?]**
>
> Thanks for this question.
>
> When there exists a stable feature that determines the label, MARG-CTRL is better than two-stage shortcut-mitigating methods in the pure setting. This stems from the fact that two-stage methods require validation group annotations for model selection and the pure setting does not provide these annotations. On the other hand, as we show in the results in the general response [[LINK]](https://openreview.net/forum?id=zyZkaqNnpa&noteId=jfFAwQVsZg) MARG-CTRL strictly outperforms two-stage methods in the pure setting.
>
> Even when the stable feature does not determine the label, MARG-CTRL mitigates shortcuts in practice: MARG-CTRL performs competitively on CelebA even though there is label noise in the dataset, meaning that the stable feature is not perfect (see figure 14 in the appendix). When the stable feature is imperfect, validation group annotations are available, and ERM-trained models help infer groups accurately, two-stage methods can outperform MARG-CTRL methods.
>
> We have added this discussion to the paper.

---

### Official Review · Reviewer_Xk5J · 2023-07-06

**Soundness:** 3 good
**Presentation:** 3 good
**Contribution:** 3 good
**Rating:** 6
**Confidence:** 3

**Summary:**

Having had my concerns addressed by the authors I have updated my score.
-----------------------------------------------------------------------------------------



* The paper proposes an explanation for why neural networks tend to learn spurious features over stable fratures which lead to a lower loss and better performance.
* The main argument for this is due to the maxmargin losses induced through cross entropy, to counter this the authors propose several losses which induce a uniform margin.
* The authors propose a nice example and demonstrate the results on vision and language datasets.

**Strengths:**

* The paper addresses a very important topic of the simplicity bias, and goes a significant way in addressing it.
* The comments and explanations are clear and concise, and follow a clear narrative.
* I find the arguments that inductive bias towards uniform margins for perception tasks to be convincing.
* The experiments are well conducted and clearly support their statements

**Weaknesses:**

* I find the scatter approach to testing many margin based losses a little un elegant, I would rather know which is the most effective in the paper and then see the results for other attempted losses in the Appendix. But this is just me preference.
* Is Corollary one explicitly followed up with an experiment? I would like to know if it actually exhibits this behaviour or is there just the potential to?
* The experimental datasets are rather limited, Waterbirds, CelebA, Wilds etc. Not a negative per se, but it would be nice to see authors trying to improve the testing framework for this probelm.

**Questions:**

* The examples in the paper tend to be restricted to linear models, which is not the case in most perception systems, do you have any intuition if these conclusions carry through to a non-linear case? I'm aware it makes the analysis harder, and I'm not asking you to perform it. But it would be nice to hear any issues that may arrise.
* Are there any experiments in the non over-parameterised setting? As this is discussed in the paper, I would have expected to see them in the experiments.
* Do you think these shortcuts are a bit artifificial? For instance in the waterbirds dataset, the background should be an important feature in the decision, but not the primary feature, does this method enable predictions from multiple features? Or do you think it will just rely on one?

**Limitations:**

Not explicitly discussed

---

> ### Author Rebuttal · Authors · 2023-08-10
>
> Thank you for your thoughtful feedback. We addressed your questions and concerns below. If any residual concerns remain, we would be glad to discuss further. If no concerns remain, we would appreciate it if you could raise your score.
>
> **[I find the scatter approach to testing many margin based losses a little un elegant]**
>
> We study multiple MARG-CTRL methods to show that the principle of controlling margins is what helps mitigate shortcuts, as opposed to properties specific to a single method.
>
> **[Is Corollary one explicitly followed up with an experiment? I would like to know if it actually exhibits this behaviour or is there just the potential to.]**
>
> Thanks for this question.
>
> Figure 1 in the pdf [[LINK]](https://openreview.net/attachment?id=jfFAwQVsZg&name=pdf) empirically demonstrates that even without overparameterization, default-ERM builds linear models that rely more on the shortcut feature than the stable feature. On the same data, MARG-CTRL builds linear models that do not rely on the shortcut feature and achieve 100% test accuracy.  See [[LINK]](https://openreview.net/forum?id=zyZkaqNnpa&noteId=IHkY6ZuSeA) for details. We have added experiments with linear models to the paper.
>
> **[The experimental datasets are rather limited. It would be nice to see authors trying to improve the testing framework for this probelm]**
>
> Thanks for this suggestion. We chose these datasets as they are amongst the most popular ones in the spurious correlations literature [1,2,3].
>
> One key contribution of this work is an evaluation framework that is more challenging than the ones used in prior work, called the pure setting. Existing methods like CnC and JTT crucially require validation group annotations for model selection to mitigate shortcuts [3,4]. However, given a stable feature that determines the label and a shortcut that does not, using the stable feature alone should achieve the highest accuracy in both classes. In turn, class labels alone should be sufficient to mitigate shortcuts. Further, for new tasks, determining shortcuts is a laborious manual process, which means shortcut annotations will often be unavailable.
>
> If one should be able to select models with validation class labels alone and unknown shortcuts can exist, model selection metrics should only use class labels. This motivates the pure setting without access to any group annotations. In the pure setting, MARG-CTRL outperforms two-stage shortcut-mitigating methods as the latter push the work to validation by using group annotations to select models.
>
> We have since updated the validation strategy in the pure setting and give the updated table of results in the general response [[LINK]](https://openreview.net/forum?id=zyZkaqNnpa&noteId=jfFAwQVsZg). We summarize the results here:
>
> - Every MARG-CTRL outperforms every two-stage shortcut mitigating method.
> - Every MARG-CTRL method outperforms ERM on every dataset.
> - In contrast, CnC and JTT without group annotations sometimes do not outperform ERM.
>
> [1] https://arxiv.org/abs/1911.08731
>
> [2] https://arxiv.org/abs/2110.14503
>
> [3] https://arxiv.org/abs/2107.09044
>
> [4] https://arxiv.org/abs/2203.01517
>
> **[The examples in the paper tend to be restricted to linear models. Do you have any intuition if these conclusions carry through to a non-linear case? I'm aware it makes the analysis harder, and I'm not asking you to perform it. But it would be nice to hear any issues that may arrise.]**
>
> Thanks for this question.
>
> The experiments in the paper empirically confirm that our theoretical insights do in fact extend to real deep neural networks: the solutions motivated by the theory — margin-control techniques — mitigate shortcuts in actual deep learning models on real-world vision and language datasets.
>
> Many works in the literature study linear models to develop insights into why deep models fail to be robust [1,2,3]. In a similar vein, we show that shortcut learning occurs in linear models despite the presence of a perfect stable feature because of max-margin classification. As default-ERM also maximizes margins in neural networks [4], the insights from our theory should extend beyond linear models which we confirm empirically.
>
> We suspect the main challenges in extending the theoretical analysis to neural networks would involve tightly lower-bounding the norm of max-margin solutions that depend only on the stable feature. The reason this may be hard is that we may also have to consider models whose first layer depends on the shortcut but that second layer combines the ReLU responses in ways that cancels out the influence of the shortcut. If one doesn’t consider these models, the norm may not be lower-bounded tightly enough to separate the max-margin predictors that only use the stable feature from the ones that rely on the shortcut.
>
> We have added this discussion to the paper.
>
> 1. https://proceedings.mlr.press/v119/sagawa20a.html
> 2. https://openreview.net/forum?id=fSTD6NFIW_b
> 3. https://openreview.net/forum?id=dQNL7Zsta3
> 4. https://openreview.net/forum?id=SJeLIgBKPS
>
> **[Does this method enable predictions from multiple features?]**
>
> MARG-CTRL only encourages the final output of models to be similar in magnitude across all samples. MARG-CTRL does not restrict which features are used.
>
> When using models like neural networks, multiple features are combined to produce a prediction. For example, in datasets like Waterbirds, multiple features together form the shape of the bird, and the shape of the bird is a stable predictor for the task of determining whether the bird is a waterbird or a landbird. MARG-CTRL relies on the available stable features to mitigate shortcuts in waterbirds in both the traditional and the pure setting. This empirically demonstrates that MARG-CTRL does enable prediction from multiple features.
>
> We have added this discussion to the paper.

---

> > ### Comment · Reviewer_Xk5J · 2023-08-15
> > **Response to Authors**
> >
> > Thank you for your response to my questions and queries, I feel they have been addressed sufficiently and have raised my score accordingly.

---

> > > ### Author Response · Authors · 2023-08-15
> > >
> > > Thank you for your consideration!

---

### Official Review · Reviewer_8MoS · 2023-07-07

**Soundness:** 3 good
**Presentation:** 2 fair
**Contribution:** 3 good
**Rating:** 6
**Confidence:** 3

**Summary:**

This paper explores the phenomenon of models using easy-to-learn spurious features (aka, shortcuts) instead of reliable but harder-to-learn true features. They find in their theoretical that max-margin relies on the spurious feature while controlling for uniforming margin induces learning the true feature. With this insight, they try numerous loss functions that provide a bias towards uniform margins and find improved worst-group accuracy on a variety of tasks.

**Strengths:**

* Experiments show good results for their methods.

**Weaknesses:**

* The informal statement of Theorem 1 in the main text is too vague in my opinion. In particular, it's not stated what the regime of $n$ is and how these other quantities interact with it.
* The graphs are tough to read (too small) and the legends overlap with the plots.

**Questions:**

* How does the dimensionality (e.g., choosing $d=100$) come into play? The story setting up the analogy does not in principle rely on the dimensionality all, just the gradient sizes of the noisy and true signals. I'm guessing it has something to do with interpolation, but I think it'd be good to have further discussion of this.
* What's the connection between the neural network trained in section 2.1 and the linear model theorems? What happens if you train a logistic regression model instead?
* Similar toy settings as in this paper (such as (Sagawa et al., 2020)) have been examined in the past and seemed to rely on overparameterization for their phenomena. What are the particulars in this paper's setup that allow us to observe phenomena in the underparameterized setting?
* How should I think of the different MARG-CTRL? What factors cause their difference in performance and how should I make the decision in which to pick?

**Limitations:**

The authors adequately addressed the limitations.

---

> ### Author Rebuttal · Authors · 2023-08-10
>
> Thank you for your thoughtful feedback. We addressed your questions and concerns below. If any residual concerns remain, we would be glad to discuss further. If no concerns remain, we would appreciate it if you could raise your score.
>
> **[The informal statement of Theorem 1 is too vague. In particular, what is the regime of n and how do other quantities interact with it.]**
>
> We have updated the paper to have a full version of theorem 1, shown below:
>
> > **Theorem 1:** Let $W^*$ be the max-margin predictor on $n$ training samples from the linear perception task in equation 1 and let the leftover group be of size $k = (1-\rho)n.$ There exist constants $C_1, C_2, N_0 > 0$ such that $$\begin{align} \forall \\, n > N_0, \quad\forall \\,\\, \rho \in \left[ 0.9, 1 - \frac{5}{n}\right], \quad \forall \\,\\, d \geq C_1 (1-\rho)n \log (3n),  \quad \forall \\,\\, B > C_2 \sqrt{\frac{d}{(1-\rho)n}}, \end{align}$$
>
> with probability at least $1-\frac{1}{3n}\\,$ over draws of the training data, it holds that $B W_z^* > W_y^*$.
>
> Briefly, the result holds for all $n > N_0 $ where $d$ scales with the size of the leftover group, up to logarithmic factors.
>
> **[Graphs too small and legends overlap.]**
>
> We have made the plots bigger and cleaned them up.
>
> **[How does the dimensionality ($d=100$) comes into play?]**
>
> The reviewer is right in that with the additional dimensions of noise, default-ERM can interpolate the training data without relying on the stable feature by using
>
> 1. The shortcut feature to classify the shortcut group
> 2. The noise to fit the leftover group, i.e. overfit to the leftover group
>
> In figure 3 in the pdf in the general response [[LINK]](https://openreview.net/attachment?id=jfFAwQVsZg&name=pdf), we show that as data dimension increases, the model achieves worse test leftover group accuracy, meaning that the model depends more on the shortcut. We train the same one-layer-network that we used in the paper on data of increasing dimension and plot its test leftover group accuracy. This larger shortcut reliance occurs because when more noise is available, the model can depend more on the shortcut while driving down leftover group loss by overfitting to it, all within the same norm budget.
>
> **[Connection between the neural network in section 2.1 and linear model theorems? What happens with logistic regression model?]**
>
> We originally ran the model with a neural network experiment to demonstrate that the shortcut learning behavior is not restricted to linear models. We have since added the linear model experiments to the paper to show that the analysis in Theorem 1 holds empirically.
>
> We provide the plots in figure 1 in the pdf [[LINK]](https://openreview.net/attachment?id=jfFAwQVsZg&name=pdf). Figure 1 empirically demonstrates that even without overparameterization, default-ERM builds linear models that rely more on the shortcut feature than the stable feature. On the same data, MARG-CTRL builds linear models that do not rely on the shortcut feature and achieve 100% test accuracy. See [[LINK]](https://openreview.net/forum?id=zyZkaqNnpa&noteId=IHkY6ZuSeA) for details.
>
> **[Similar toy settings from the literature (such as (Sagawa et al., 2020)) assume overparameterization for their phenomena. What in this paper's setup allows us to observe phenomena in the underparameterized setting?]**
>
> Existing results like Sagawa et al. typically assume overparameterization to ensure linear separability of the data and then rely on results like Soudry et al. (https://arxiv.org/pdf/1710.10345.pdf) that says default-ERM on linear models converges in direction to the max-margin predictor. In our setting, the stable feature determines the label which guarantees linearly separability without assuming overparameterization.
>
> We then perform a tight analysis to ensure that the covariate dimension required to induce shortcut learning only scales linearly in the size of the leftover group, up to logarithmic factors in sample size. Given this scaling relationship, we can show that even underparameterized models exhibit shortcut learning when the leftover groups are small enough.
>
> Our work focuses on shortcut learning in default-ERM in the scenario where the stable feature determines the label, which means the bayes-optimal predictor on the training distribution only relies on the stable feature. Understanding why default-ERM favors shortcut-based classification instead of its target bayes-optimal predictor supports discovering novel strategies to mitigate shortcuts — like MARG-CTRL. Further, such understanding can help identify tasks where one should challenge default-ERM as the primary choice for modeling.
>
> We have added this discussion to the paper.
>
> **[How should I think of the different MARG-CTRL? What factors cause their difference in performance and how should I make the decision in which to pick?]**
>
> Great question. Two factors affect the performance of MARG-CTRL method:
>
> - the hyperparameter like the temperature in sigmoid-damping.
>
>    - Using too large a temperature makes MARG-CTRL too similar to log-loss and uniform margins are not encouraged, which fails to mitigate shortcuts. Using too small a temperature can mean optimization takes a long time to converge. Searching over a finer grid for the hyperparameters strikes a good tradeoff between optimization speed and controlling margins, and make different MARG-CTRL methods perform more similarly.
>
> - the penalty each method places on small margins and misclassifications compared to the penalty on large margins.
>
>     - For example, figure 11 in the paper shows that sigmoid-stitching penalizes large margins more than sigmoid-damping. In turn, sigmoid-damping may favor fewer small margins.
>
>
> In practice, we recommend using one MARG-CTRL method and tuning the hyperparameters: while we do not have theoretical guarantees. MARG-LOG is usually near the best and is a safe bet. Studying the existence of uniformly optimal MARG-CTRL would be fruitful future work.

---

> > ### Comment · Reviewer_8MoS · 2023-08-19
> > **Reply to authors**
> >
> > Thank you for addressing my concerns, I will raise my score.

---

> > > ### Author Response · Authors · 2023-08-19
> > >
> > > Thank you for your feedback and consideration!

---

### Author Response · Authors · 2023-08-10
**Rebuttal details**

**[Discussion of the updated model selection metric and results in the pure setting]**

The original model selection/validation strategy in the pure setting used the worst class error. We noticed that the worst class can sometimes be the class that does not contain the worst group, making this validation insensitive to certain groups. Instead, we now validate with **label-balanced accuracy** to make sure neither class is removed from consideration in model selection. This validation metric still only uses the class labels.

We repeat the summary of the updated results here:

- Every MARG-CTRL outperforms every two-stage shortcut mitigating method.
- Every MARG-CTRL method outperforms ERM on every dataset.
- In contrast, CnC and JTT without group annotations sometimes do not outperform ERM.

Results table in the pure setting. We bold the MARG-CTRL methods.
|               | CelebA       | WB           | Civil        |
|---------------|--------------|--------------|--------------|
| ERM           | 57.5 ± 5.8   | 69.1 ± 2.1   | 60.7 ± 1.5   |
| CNC           | 67.8 ± 0.6   | 60.0 ± 8.0   | 61.4 ± 1.9   |
| JTT           | 53.3 ± 3.3   | 71.7 ± 4.0   | 53.4 ± 2.1   |
|||||
| **marg-log**     | 74.2 ± 1.4   | 77.9 ± 0.3   | 66.8 ± 0.2   |
| **norm**          | 74.2 ± 1.9   | 74.3 ± 0.1   | 63.9 ± 3.2   |
| **$\sigma$-damp**       | 70.8 ± 0.3   | 74.8 ± 1.6   | 65.6 ± 0.2   |
| **sd**           | 70.3 ± 0.3   | 78.7 ± 1.4   | 67.8 ± 1.3   |
| **$\sigma$-stitch**        | 76.7 ± 0.6   | 74.5 ± 1.2   | 66.0 ± 1.0   |


These results showcase the value of using MARG-CTRL over the more expensive two-stage mitigating methods.

The pure setting that we introduce is a new and challenging testing framework that does not provide any groups or shortcut information in training or validation.

**[Details of training an underparameterized linear model with default-ERM and MARG-CTRL]**

We trained a linear model on data from the linear perception task (from eq 1) which we repeat here:

$$\begin{align}Y \sim \textrm{Rad},
  \quad
Z = & \quad Y \text{ with probability }
\rho
  \qquad   \qquad \delta \sim \mathcal{N}(0, \mathbf{I}^{d-2}),
 \quad
  X
  = \left[B*Z ,Y, \delta \right].
\\\\
= & - Y \text{ with probability } 1-\rho
\end{align}$$

- With $d=300, n=1000, B=10$, the training data comes from $p_{\rho=0.9}$ and the test data comes from $p_{\rho=0.1}$.
- The shortcut’s relationship with the label flips between train and test, due to $\rho$ changing from 0.9 to 0.1.
- A linear model that depends more on the stable feature than the shortcut feature — that is the model output does not flip sign when the shortcut feature flips — achieves better-than-random (>50\\%) test accuracy. For technical details, see below:

   - For a linear model $W$, with parameters $[W_z, W_y, W_e]$ for the shortcut, stable feature, and noise respectively, the linear model margin on a sample is $$yW^\top x = W_y + B y z W_z + y W_e^\top \mathbf{\delta}$$

   - When the shortcut does not match the label, the margin is $W_y - B W_z + y W_e^\top \mathbf{\delta}$
   - When the linear model depends more on the stable feature, i.e. $W_y > B W_z$, we have that $W_y - B W_z > 0$
   - Noting that $W_e^\top \mathbf{\delta}$ is zero-mean gaussian noise, the probability of the sign of the margin $yW^\top X$ flipping to negative is the probability of zero-mean gaussian variable being smaller than a negative number, which is $<0.5$
   - Thus, the probability of error on such samples is $<0.5$, so accuracy is $>50\\%$.
   - The margin has a larger mean when the shortcut matches the label, so the accuracy is also $>50\\%$.

- However, the plot shows that default-ERM builds a linear model with $<40\\%$ test accuracy which is worse-than-random ($50\\%$).
- This shows that the non-overparameterized linear model trained by default-ERM depends more on the shortcut feature than the stable feature.

As opposed to default-ERM, using MARG-CTRL mitigates shortcut learning and achieves perfect test accuracy. We plot accuracy and loss curves with sigmoid-damping and sigmoid-stitching in figure 2 in the pdf [[LINK]](https://openreview.net/attachment?id=jfFAwQVsZg&name=pdf), showing that the two methods yield linear models with similar test losses in the shortcut and leftover groups. Depending on the shortcut makes the margins different in the two groups, meaning that the model relies solely on the stable feature to achieve $100\\%$ test accuracy.

---

### Author Rebuttal · Authors · 2023-08-10


# General response

We thank the reviewers for their thoughtful feedback.  We are glad the reviewers find that
- Our paper is well-written
  - “The comments and explanations are clear and concise, and follow a clear narrative” - Xk5J
  - “The paper is well-written and easy to understand” - PgqC
- Our paper presents a thorough, novel, and significant analysis
  - “This paper presents a thorough analysis of the shortcut learning problem” - PgqC
  - “I have not seen an analysis from this perspective before.” - 9QKU:
  - “The paper addresses a very important topic of the simplicity bias, and goes a significant way in addressing it.” - Xk5J
- Our solutions are novel and effective and our experiments are well-conducted
  - “The authors propose a novel solution, MARG-CTRL, which has been shown to effectively mitigate shortcut learning in several tasks.” - PgqC
  - “Experiments show good results for their methods.” - 8MoS
  - “The experiments are well conducted and clearly support their statements” - Xk5J


Our paper begins with the premise that given a stable feature that determines the label, a model trained with gradient-based optimization of the log-loss, which we call default-ERM, can achieve zero training loss using only the stable feature. This model that only uses the stable feature always generalizes outside the training distribution.

However, in popular tasks like classifying hair color from images of celebrities, default-ERM builds a model that relies on shortcuts even though the stable feature is perfect in that it determines the label. Models that rely on shortcuts do not generalize outside the training distribution.

Why does default-ERM prefer shortcuts despite the presence of a perfect stable feature?


**[Main contributions]**

1. We prove that default-ERM’s inductive bias toward maximizing margins causes shortcut learning in linear models even without overparameterization.
2. In contrast, we show that an inductive bias towards uniform margins favors perfect stable features.
3. We develop a family of methods that encourage uniform margins; we call these margin-control (MARG-CTRL) methods.
4. We demonstrate that MARG-CTRL mitigates shortcuts as well or better than the best of default-ERM and existing two-stage shortcut-mitigating methods, in deep models for vision and language tasks.
5. We introduce a new experimental setting without group annotations in training or validation. In this more challenging setting, MARG-CTRL strictly outperforms both default-ERM and the more computationally expensive two-stage shortcut-mitigating methods.

**[Paper improvements made in response to feedback]**

In response to the reviewers' feedback, we have improved the paper as follows:

- We have added experiments with underparameterized linear models with both default-ERM and MARG-CTRL to empirically validate theorems 1 and 2, and corollary 1. See pdf [[LINK]](https://openreview.net/attachment?id=jfFAwQVsZg&name=pdf). (reviewers 8MoS, Xk5J)
- We have expanded the discussion of the limitations and trade-offs of MARG-CTRL (reviewers PgqC, Xk5J).
- We have included additional discussion about theorem 1, elaborating on the roles of dimension and the scalar factor $B$ in inducing shortcut learning. (reviewer 9QKU)
- We altered the model selection strategy in the pure setting where group annotations are absent (for details see below [[LINK]](https://openreview.net/forum?id=zyZkaqNnpa&noteId=IHkY6ZuSeA)).  This alteration improves the test performance of multiple methods, including some MARG-CTRL methods and some non-MARG-CTRL methods. We summarize the updated results here:
  - Every MARG-CTRL outperforms every two-stage shortcut mitigating method.
  - Every MARG-CTRL method outperforms ERM on every dataset.
  - In contrast, CnC and JTT without group annotations sometimes do not outperform ERM.

	(reviewer Xk5J)

**[Updated results in the pure setting]**

|               | CelebA       | WB           | Civil        |
|---------------|--------------|--------------|--------------|
| ERM           | 57.5 ± 5.8   | 69.1 ± 2.1   | 60.7 ± 1.5   |
| CNC           | 67.8 ± 0.6   | 60.0 ± 8.0   | 61.4 ± 1.9   |
| JTT           | 53.3 ± 3.3   | 71.7 ± 4.0   | 53.4 ± 2.1   |
|||||
| **marg-log**     | 74.2 ± 1.4   | 77.9 ± 0.3   | 66.8 ± 0.2   |
| **norm**          | 74.2 ± 1.9   | 74.3 ± 0.1   | 63.9 ± 3.2   |
| **$\sigma$-damp**       | 70.8 ± 0.3   | 74.8 ± 1.6   | 65.6 ± 0.2   |
| **sd**           | 70.3 ± 0.3   | 78.7 ± 1.4   | 67.8 ± 1.3   |
| **$\sigma$-stitch**        | 76.7 ± 0.6   | 74.5 ± 1.2   | 66.0 ± 1.0   |

-------

We address one common concern here.

**[Does the theory that focuses on linear models and linearly separable data provide insights into shortcut learning with neural networks on general datasets?]**


Reviewers 9QKU and Xk5J raised concerns that the theory is restricted to linear models. They asked questions about whether the insights can be extended to non-linear models.

The experiments in the paper empirically confirm that our theoretical insights do in fact extend to real deep neural networks: the solutions motivated by the theory — margin-control techniques — mitigate shortcuts in actual deep learning models on real-world vision and language datasets.

Many works study linear models to develop insights into why deep models fail to be robust  [1,2,3]. In a similar vein, we show that shortcut learning occurs in linear models despite the presence of a perfect stable feature because of max-margin classification. As default-ERM also maximizes margins in neural networks [4], the insights from our theory should extend beyond linear models, which we confirm empirically.


1. https://proceedings.mlr.press/v119/sagawa20a.html
2. https://openreview.net/forum?id=fSTD6NFIW_b
3. https://openreview.net/forum?id=dQNL7Zsta3
4. https://openreview.net/forum?id=SJeLIgBKPS

---

### Decision · Program_Chairs · 2023-09-21

**Decision:**

Accept (poster)

**Comment:**

This paper theoretically analyzes shortcut learning, showing that max-margin learning leads to spurious feature learning, and use this to design loss functions which encourage uniform margins, finding improved worst-group accuracy. Reviewers found the question interesting, paper well-written, the theory well done, and the proposed solutions effective. I therefore recommend acceptance.